# On understanding mountainous carbonate basins of the Mediterranean using parsimonious modeling solutions

Shima Azimi[1,2], Christian Massari[2], Giuseppe Formetta[1], Silvia Barbetta[2], Alberto Tazioli[3], Davide Fronzi[3], Sara Modanesi[2], Angelica Tarpanelli[2], and Riccardo Rigon[1]

[1]University of Trento, Department of Civil, Environmental and Mechanical Engineering, Center Agriculture, Food and Environment (C3A), Trento, Italy
[2]National Research Council (CNR), Research Institute for Geo-Hydrological Protection, Perugia, Italy
[3]Università Politecnica delle Marche, Department of SIMAU, Ancona, Italy

**Correspondence:** Riccardo Rigon (Riccardo.rigon@unitn.it)

**Abstract.**

The study aims to demonstrate that an effective solution can be implemented for modeling complex carbonate basins, in the situation of limited data availability. Considering the alternative modeling approaches under circumstances of data shortage is more significant knowing the vulnerability and effectiveness of these kinds of basins to drought and climate change conditions. In this regard, a hybrid approach that combines time series analysis and reservoir modeling is proposed to describe behaviour in carbonate basins. Time series analysis estimates the contributing area and response time of the fractured carbonate system beyond the catchment's hydrographic boundaries. The obtained results align with previous literature-based field surveys. This information is then used to develop a conceptual reservoir system using the GEOframe modeling system. The model is validated using in situ discharge observations and Earth Observations (EO) data on evapotranspiration and snow. Model reliability is assessed using traditional goodness of fit indicators, hydrological signatures, and a novel statistical method based on empirical conditional probability. This approach enables detailed analysis and investigation of water budget components in Mediterranean carbonate catchments, highlighting their response to significant precipitation deficits.

Overall, our results demonstrate that flows from carbonate rock areas outside the hydrographic boundaries significantly impact the water budget of the upper Nera River. The storage capacity of the carbonate basin plays a crucial role in sustaining river discharge during drought years. In a single dry year, meteorological drought is considerably attenuated, while in subsequent dry years, it is slightly intensified. Multi-year droughts result in slower recovery due to the time required for precipitation to replenish the depleted storage that supported river discharge in previous dry years. This unique behavior makes these basins particularly vulnerable to more severe and frequent drought episodes expected under future climate change.

Keywords: *Karst; External groundwater discharge; Parsimonious hydrological modeling; Empirical conditional probability-based bias correction; Droughts.*

# 1 Introduction

Carbonate/karst landscapes represent approximately 7-12 percent of the Earth's continental area and they provide a significant challenge for hydrologists (Hartmann et al., 2014). Due to the capability of these landscapes to retain water for a longer period (i.e., long-term hydrological memory catchments), their storage plays an important role in the control of drought propagation and delayed hydrological recovery (Alvarez-Garreton et al., 2021).

Generally, a carbonate/karst landscape forms when the percolated precipitation dissolves the subterranean carbonate bedrock and creates extensive fissures, open fractures, conduits, and caves. This can result in a complex network of groundwater flowpaths occurring within the same or adjacent aquifers (Kiraly et al., 1995). To model these types of systems one powerful solution is to use distributed, process-based models (PB) (e.g., Rooji, 2020; Hartmann et al., 2014), which are based on solvers for groundwater partial differential equation. Yet, the main challenge of this kind of distributed model is that they require a large amount of hydrogeological data and extensive field analysis to set appropriate physical parameter values and correct boundary conditions. On top of that, large computational power is needed to run these models (Li et al., 2022).

An alternative to PB is black-box models based on machine learning (MLM) in which all the details about the structure of the aquifer and the hydrodynamics parameters are not needed, e.g., Tapoglou et al. (2014); Castilla-Rho et al. (2015). Although the implementation of MLM is easy, their model parameters do not have a physical meaning and are only indirectly related to the characteristics of the carbonate system (Zhou et al., 2019). Furthermore, MLM does not explicitly solve the water budget and thus it is not possible to have information about the dynamics of all water budget components.

Hydrological Dynamical Systems (lumped models, HDSys) represent another type of model, based on a set of ordinary differential equations (ODEs) that conceptualize the entire carbonate system as a series of reservoirs (e.g., Bancheri et al., 2019; Hartmann et al., 2014; Rimmer and Hartmann, 2012; Butscher and Huggenberger, 2008; Tritz et al., 2011; Jukic and Denić-Jukić, 2009; Duboisl et al., 2020). Instead of explicitly considering spatial variables, HDSys specify the interconnection of fluxes between different reservoirs, which leads to reducing the computational complexity. However, HDSys still require the definition of model parameters, which typically rely on calibration and inverse modeling using monitored discharge data or other relevant data sources (Hartmann et al., 2014). Several studies have also explored modeling the fast and slow drainage from carbonate systems using tracer information (e.g., Rimmer and Hartmann, 2012; Dubois et al., 2020). This approach involves introducing an artificial tracer into a sinkhole and then tracking the tracer's movement in the surrounding areas at different times (Hartmann et al., 2014; Zhang et al., 2021; Nanni et al., 2020). While this technique can be useful, it is time-consuming and may not always be feasible due to accessibility issues.

These HDSys can be conjugated by techniques that rely on the correlation between precipitation and discharge can provide valuable insights about the behavior of carbonate systems, particularly in situations where field information about water circulation is limited. It could also provide useful information in the situation of missing tracer test analysis. For example, Fiorillo and Doglioni (2010) used cross-correlation analysis to estimate the time that water requires to flow through fissured aquifers. Another useful method, borrowed from applied economics (Kristoufek, 2014, 2015), was employed by Giani et al. (2021) to estimate the basin response time of hydrographs to precipitation, with successful results. However, according to the authors'

best knowledge, to date, this data analysis technique has not been applied to complex carbonate systems to determine their hydrological response to precipitation.

This study aims to address the following five research questions (RQs):

1. Can the complex response of carbonate catchments to precipitation be modeled with HDSys relying only upon streamflow and precipitation time series, aided by cross-correlation analysis?

2. What type of modeling solution is suitable for this task, and is a parsimonious modeling approach appropriate?

3. Are the classic goodness of fit scores enough to evaluate the reliability of the models?

4. What is the impact of external contributing areas on streamflow in catchments with fractured carbonate rocks? To what extent does this contributing area affect the total streamflow from small headwater catchments to the main outlet?

5. What is the role of storage in sustaining streamflow during the years with significant precipitation deficit in these types of catchments?

We have examined the water budget of the Nera River basin, which is a significant tributary of the Tiber River, the second-largest river in Italy. The Nera River basin contributes nearly 50% of the total discharge of the Tiber River and is characterized by a significant portion of fissured and fractured carbonate rocks feeding the river discharge by releasing a large amount of groundwater into the river bed from streambed springs. Thus, this catchment is a good representative of the carbonate catchments for answering the RQs. Additionally, groundwater data shortage is a problem that is not unique to the Upper Nera River area, and the findings of this study could help inform water management and policy decisions in other carbonate basins as well. By providing a comprehensive analysis of the water cycle in this area, this study could also help identify potential sources of water stress and suggest strategies to mitigate them.

## 2   Study Area and datasets

### 2.1   Study area

The Nera River is the largest tributary of the Tiber River and its sources are in the Sibylline Mountains in central Italy. It is 116 kilometers long and flows almost entirely in a deep valley called Valnerina, through limestone formations that constitute huge aquifers that are drained by the river. The landscape is mainly hilly and mountainous and is almost totally covered by forests, with pastures at higher elevations (from 1200 to 2200 m a.s.l). The Upper Nera River basin up to the Visso River station (our study area) covers an area of around 110 km$^2$ and the elevation ranges between 570 and 2200 m a.s.l., with a mean basin slope of 25 %. The basin is characterized by a Mediterranean climate, with precipitation concentrated mostly in the autumn-spring period, when floods generally occur. The annual precipitation and the average temperature of the basin are around 1100 mm and 10 °C, respectively.

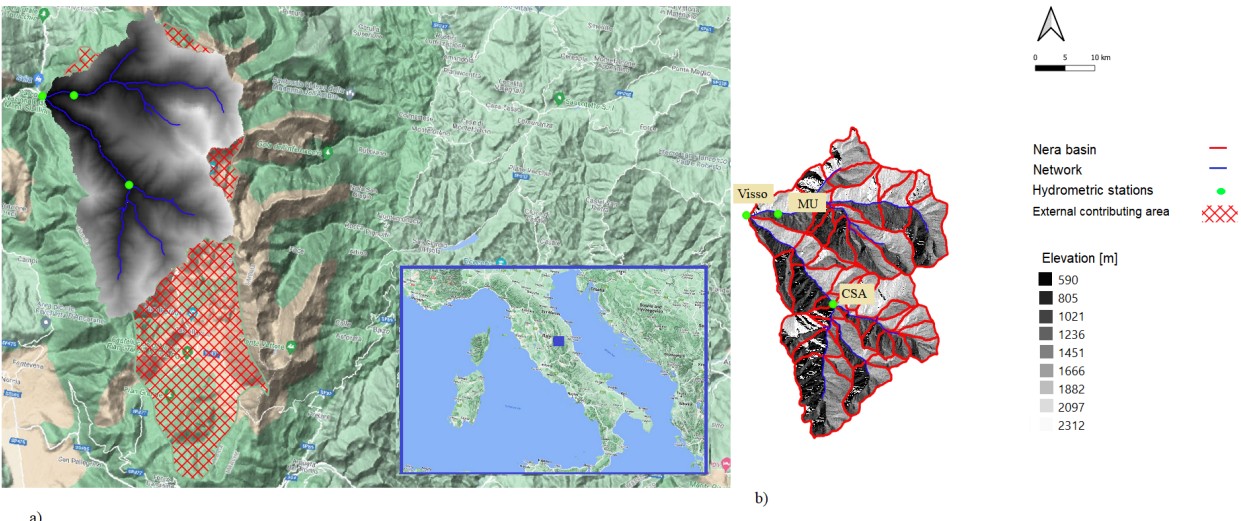

**Figure 1.** (a) The study area with the DEM of the basin, additional external area, and the locations of the hydrometric stations on the study area. (b) Delineation of the Hydrologic Response Units (HRUs) in the hydrological basin, based on the generic method (shape and morphology of the basin). The background map has been retrieved from Google Satellite Hybrid (© OpenStreetMap).

The discharge of the Nera River is contributed to by a set of permanent streambed springs fed by large limestone aquifers, already studied in Boni et al. (1986), that give rise to complex groundwater-surface water interactions. Nanni et al. (2020) and Mastrorillo et al. (2019) showed how these aquifers extend beyond the limits of the river basin into the wide and complex hydrogeological boundary of the Sibylline Mountains. Mastrorillo et al. (2019) estimated that the total contributing area of the fractured carbonate system outside the hydrographic boundaries of the basin (our study area) is around 97 km$^2$. Fronzi et al. (2021) did several tracer tests and showed that the river is fed (from the southeast of the basin) by carbonate aquifers with an area almost 4 times larger than the one enclosed by the river station of Castelsantangelo, located upstream of Visso (Fig. 1(a)). Similar findings were found for the Ussita River, the main tributary of the Nera River at Visso, which is characterized by a real contributing area almost twice that of its hydrographic boundaries (Mastrorillo et al., 2019).

## 2.2 Terrain data and ground Meteorological network

The terrain data for the geomorphological analysis of the basin were provided by the Marche Region Authority. The Horton Machine Toolbox Abera et al. (2017) was used to define the basin and the hydrographic boundaries shown in (Fig. 1(a)); the basin was then further subdivided into 47 Hydrologic Response Units (HRUs) (Fig. 1(b)).

In the study, we used the meteorological network of the area provided by the Marche Regional Authority from which we selected 32 precipitation gauges, 21 thermometers, and 3 hydrometric stations that are distributed throughout the basin (these data are provided in the supplemental material). The monitoring network provides 15-minute data for which a quality check

**Table 1.** Areas obtained from the classical hydrographic catchment delineation and from the hydrogeological survey at the outlet Visso, Castelsantangelo, and MU hydrometric stations.

| Station | River | Geom. basin area | External contributing area | Total area | Record |
|---|---|---|---|---|---|
| - | - | [km$^2$] | [km$^2$] | [km$^2$] | [yr] |
| Castelsangelo (CSA) | Nera | 17 | 70 | 87 | 2009-2017 |
| Visso | Nera | 97 | 110 | 207 | 2007- |
| Madonna dell'Uccelletto (MU) | Ussita | 45 | 40 | 85 | 2017- |

to remove anomalous values and a re-sampling to the hourly resolution were performed. Streamflow data were calculated by transforming water levels measured at the hydrometric stations via rating curves updated bi-yearly (Fig. 2). The stations used in the study are Visso and Castelsantangelo (CSA) on the Nera River, and Madonna dell'Uccelletto (MU) on the Ussita River. To prevent any confusion, from this point forward in the text, "MU" will be used to refer to the station, while "Ussita" will be used to denote the river basin.

The Visso station is the main outlet section of the study area and its data, covering the period 2007-2021, were used for the hydrological analysis. CSA station, with data available in the period of 2010-2016, is located upstream of the basin and is affected by a significant proportion of groundwater discharge coming from the external carbonate area (Fronzi et al., 2021). It should be mentioned that the continental deposits preserve this carbonate aquifer from the direct dissolution processes limiting the mature karst development in the saturated zones (Petitta et al., 2022). So the basin is not considered a fully-karst system. The two stations were affected by the seismic sequences of 2016-2018 and thus these data have been excluded from the analyses. The earthquake altered the groundwater contribution of the fractured system, determining an abrupt and sustained change in the river and spring discharges in several parts of the basin (Di Matteo et al., 2020, 2021a). The MU hydrometric station on the Ussita River is characterized by about 3 years of hourly data since Nov. 2018.

Table 1 summarizes the total contributing area (from the basin and from outside the hydrographic boundaries of the basin) of the three hydrometric stations (based on the literature ). These basin areas were derived from the terrain analysis and literature and can be estimated to be equal to 207 km$^2$ for Visso, 85 km$^2$ for MU, and 87 km$^2$ for CSA (see Fig. 1(a) and Table 1).

### 2.3 Remote sensing data

Remote sensing data of evapotranspiration (ET) and snow depth were also used to complement the validation. In particular, we used MODIS actual ET (Mu et al., 2013) and the Sentinel-1 snow depth (Lievens et al., 2019). The global MODIS ET dataset (MOD16A2/MYD16A2) provides evaporation from wet and moist soil, evaporation from rainwater intercepted by the canopy before it reaches the ground, and transpiration through stomata on plant leaves and stems with 500m spatial and 8-day temporal resolutions (Mu et al., 2013). The dataset can be downloaded from (http://files.ntsg.umt.edu/data/NTSG_Products/MOD16/).

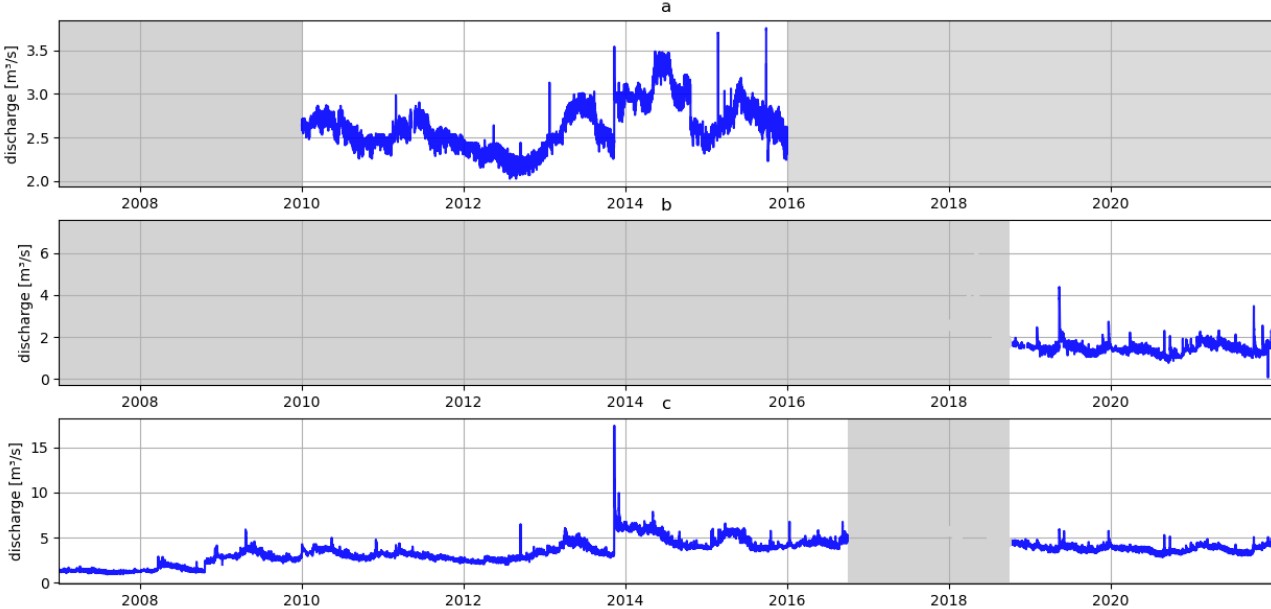

**Figure 2.** The discharge time series at the outlet of (a) Castelsantangelo (CSA), (b) Madonna dell'Uccelletto (MU), and (c) Visso (outlet) stations. The discharge values are not available for the gray color periods and have been eliminated from the hydrological analysis. All the time series are hourly.

The Sentinel-1 snow depth retrieval algorithm is based on an empirical change-detection method applied to the measurements of the cross-polarization ratio ($\sigma^0_{vh}/\sigma^0_{vv}$; in dB) (Lievens et al., 2019). The Sentinel-1 snow depth retrievals are available online at https://ees.kuleuven.be/project/c-snow.

## 3   Characterization of the basin

Data availability plays an important role in selecting a suitable modeling approach for carbonate basins. Prior to determining the groundwater contribution to river discharge with respect to the other components of the water balance, we analyzed the precipitation and streamflow time series by focusing on the area derived from the terrain analysis (i.e. without external area contribution, Table 1). Fig. 3(a1) shows a comparison of the cumulative precipitation against cumulative river discharge observed at CSA. It can be seen that cumulative river discharge is much higher than the precipitation. In particular, Fig. 3(a2) shows that the runoff coefficient of the basin at CSA – obtained by dividing the discharge at CSA by the precipitation time series – ranges between 4 and 5. This means that, assuming as the null hypothesis that the mean precipitation is constant in the red-shaded area of Fig. 1, the external groundwater contributing to the CSA outlet is at least 4 or 5 times larger than the extension of the basin delineated by terrain analysis, which is in line with Fronzi et al. (2021). Fig. 3(b1) and (b2) also show evidence of external groundwater contributions for MU, determining a runoff coefficient higher than one. The relative contri-

bution of external groundwater to river discharge tends to reduce by moving downstream, as manifested by the smaller runoff coefficient observed at Visso (Fig. 3(c1) and (c2)). The area of CSA, Ussita, and Visso increases in order, with CSA being the smallest and Visso being the largest which encompasses both CSA and Ussita. Fig. 3 illustrates that the contribution from the carbonate (red) catchment decreases with increasing catchment size, which is a reasonable expectation. To further clarification about the lower runoff coefficient of Visso, the readers could refer to Section 5.2 and Fig. 12. Note that the panels of Fig. 3(a1, b1, and c1) are obtained using the closest rain gauges to the river stations; the plots related to other rain gauges located inside and outside the analyzed basins are available in the Supplementary material (Fig. 1S & 2S).

After understanding that the external contribution to the basin is significantly greater than that provided by terrain analysis, it becomes necessary to determine the time delay between precipitation input and groundwater release into the basin.

To accomplish this, the approach proposed by Giani et al. (2021), which characterizes the time lag at which precipitation and streamflow are better correlated, is employed. To the best of the authors' knowledge, this method has not previously been used in the literature to estimate the travel time of groundwater in reservoir-based models.

Fig. 4 shows the application of the method of Giani et al. for streamflow at the CSA and MU stations and the precipitation over the basin to understand the time lag at which these two variables are most strongly correlated. Fig. 4(a) shows that $\sim$ 30-day (700 hours) is the time window at which the precipitation and discharge of CSA are most correlated, which is in line with the results of Nanni et al. (2020), who showed that the mean tracer transit time (days) is around 26(+/-3) days for CSA. We attribute the high correlation of the first time window (3 days/70 hours) to the subsurface river discharge response and that of the second to the groundwater contribution to the total river discharge. Tracer tests conducted later by Fronzi et al. (2021) demonstrated that springs emerge along the Nera River Valley and feed the river directly, as well as some streambed springs that emerge a few kilometers downstream. For MU, Fig. 4(b) shows that 167 days (4000 hours) is the window length in which the discharge and the precipitation are most correlated. Evidence of a delayed external groundwater contribution in the Ussita River discharge is also present in Mastrorillo et al. (2019), who showed that a fraction of the total external area contribution to the Nera River feeds the Ussita River basin before MU (see Table 1).

# 4 Methods for modeling basins with external groundwater contribution

Generally speaking, the topography and the derived shape of the basin give a guide on how to separate the pathways of water and obtain the precipitation volumes for each sub-basin. This general approach is evidently not applicable in the case of the Upper Nera basin, as also evidenced by the values of the runoff coefficients in section 3. However, according to the field survey and measurements conducted by Mastrorillo et al. (2009) and Nanni et al. (2020), the extension of the contributing basin area has been determined but the groundwater pathways are still unknown and this poses important challenges for models based on fully coupled 3D or 2D surface-groundwater contributions.

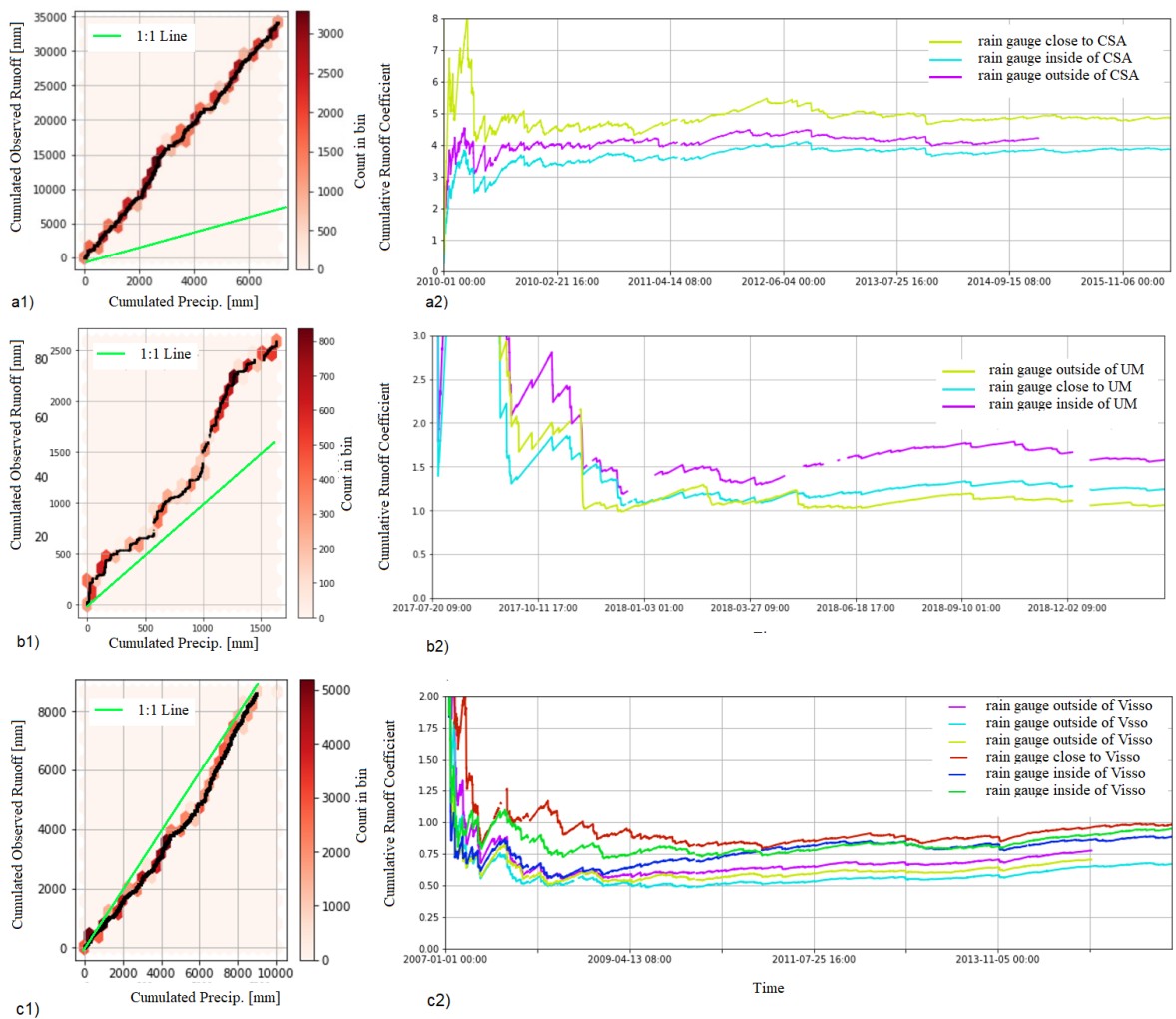

**Figure 3.** The relationship between cumulative observed discharge and cumulative precipitation at three different locations: Castelsantangelo (CSA) (panel a), Madonna dell'Uccelletto (MU) (panel b), and Visso (panel c). For each panel, the two plots are (1) cumulative observed discharge versus cumulative precipitation recorded at the closest station, the green line representing a 1:1 relationship between discharge and precipitation, and (2) the runoff coefficient time series computed by dividing the discharge by the precipitation time series recorded at different stations. The runoff coefficient varies considerably in the basin. For Castelsantangelo (CSA) (panel a) and Madonna dell'Uccelletto (MU) (panel b) the runoff coefficient is approximately 4 and 1.5, respectively, while at the outlet of the basin (Visso, panel c), the coefficient is around 1.

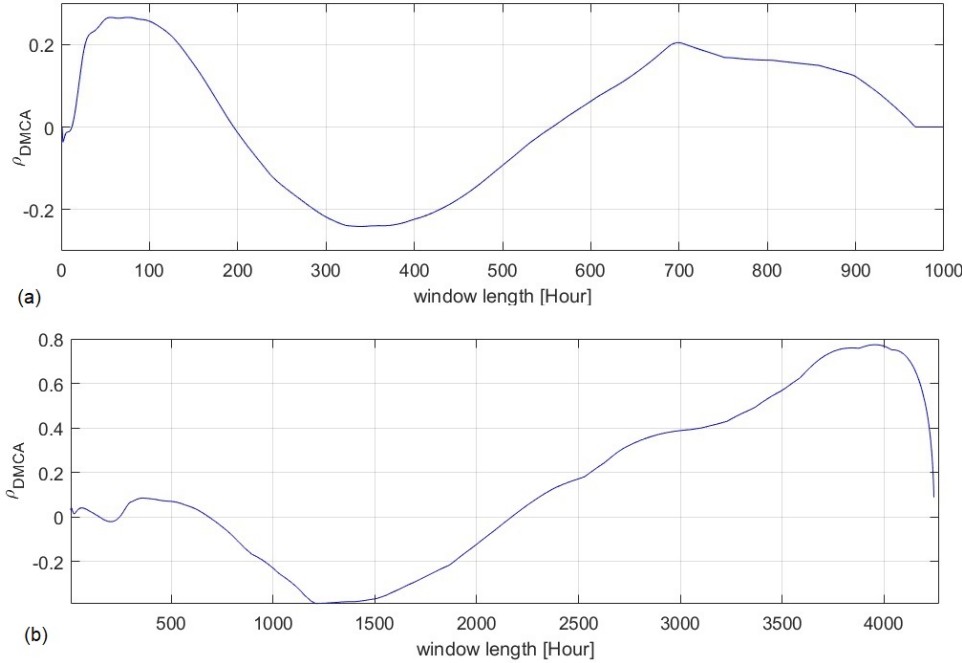

**Figure 4.** The correlation between precipitation time series and discharge for (a) Castelsantangelo (CSA) and (b) Madonna dell'Uccelletto (MU), using different time window lengths. For Castelsantangelo (CSA), the strongest correlation is related to the 3-day (around 70 hours) and 30-day (700 hours) time lags. The earlier correlation is likely linked to the surface basin's response to the precipitation, while the latter is associated with water recharge from fissured rocks. For Madonna dell'Uccelletto (MU), the highest correlation between precipitation and discharge is at 167-day (around 4000 hours) time lags. The higher correlation is likely associated with the slow response of the aquifer which responds slowly to precipitation events.

## 4.1 Model structure

To model the fissured systems we split the basin area into two main parts: the surface catchment (SC) and the "external aquifer" catchment (AC). As mentioned earlier, the SC has been partitioned into the 47 HRUs, while the ACs are considered as unique sub-basins where the water flows into the SC from the CSA and MU.

The snow contribution is taken into account for both SC and AC, as shown by the Extended Petri Net (EPN) in Fig. 5. In particular, the EPN shows how precipitation flux is partitioned into snowfall and rainfall, according to the air temperature (Formetta et al., 2014). Snow then melts, increasing the liquid water in snow, and thereafter it can either refreeze or flow and be transferred to the vegetation reservoir. Afterward, the input flux into the canopy reservoir either evaporates from the canopy or falls through to the soil (Fig. 5(b)). The part that falls to the soil is partitioned into three reservoirs, presented in Fig. 5(c). For more details on the functionality of the reservoirs, the reader is referred to the main references such as Formetta et al. (2014); Bancheri et al. (2019). To compute actual evapotranspiration (ET), we rely upon the Prospero model (Bottazzi et al., 2021),

| Snow Reservoirs | | |
| --- | --- | --- |
| Symbol | Name | Unit |
| $M$ | Melted Water | $[L\,T^{-1}]$ |
| $M^d$ | Runoff Melted water | $[L\,T^{-1}]$ |
| $P$ | Precipitation | $[L\,T^{-1}]$ |
| $P_r$ | Liquid Precipitation | $[L\,T^{-1}]$ |
| $P_s$ | Snowfall | $[L\,T^{-1}]$ |
| $R$ | Refrozen Water Rate | $[L\,T^{-1}]$ |
| $SWE$ | Snow Water Equivalent | $[L]$ |
| $T$ | Temperature | $[K]$ |
| $W_s$ | Liquid Water in snow | $[L]$ |

| Canopy Reservoirs | | |
| --- | --- | --- |
| $ETc$ | Evaporation from wetted canopy | $[L\,T^{-1}]$ |
| $Md$ | Runoff Melted water | $[L\,T^{-1}]$ |
| $Tr$ | Throughfall | $[L\,T^{-1}]$ |
| $U_1$ | Fast Runoff | $[L\,T^{-1}]$ |
| $U_2$ | Slow Runoff | $[L\,T^{-1}]$ |

| Runoff Reservoirs in non-karst area | | |
| --- | --- | --- |
| $ET_{RZ}$ | Evapotranspiration | $[L\,T^{-1}]$ |
| $Q_R$ | Surface Water Runoff | $[L\,T^{-1}]$ |
| $Q_{WG}$ | Groundwater contribution to the surface runoff | $[L\,T^{-1}]$ |
| $R_E$ | Recharge | $[L\,T^{-1}]$ |
| $S_R$ | Surface Water | $[L]$ |
| $S_{RZ}$ | Root Zone Storage | $[L]$ |
| $S_{WG}$ | Groundwater | $[L]$ |
| $T_r$ | Throughfall | $[L\,T^{-1}]$ |
| $U_1$ | Fast Runoff | $[L\,T^{-1}]$ |
| $U_2$ | Slow Runoff | $[L\,T^{-1}]$ |

| Runoff Reservoir in karst area | | |
| --- | --- | --- |
| $k$ | Mean residence time | $[T]$ |
| $Q_{low}$ | Discharge out of the karst area | $[L\,T^{-1}]$ |
| $S_{low}$ | Storage in karst area | $[L]$ |

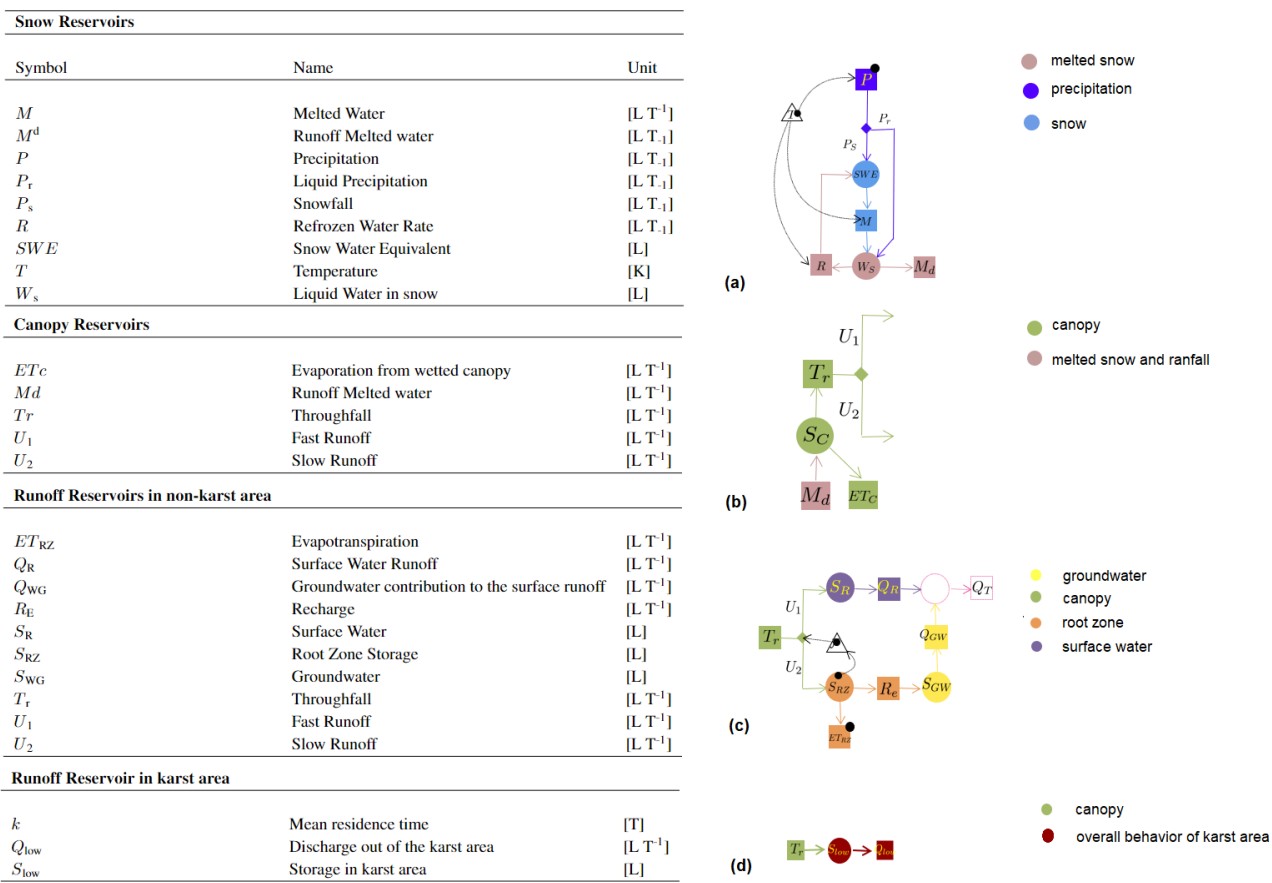

**Figure 5.** The expressions and Extended Petri Net (EPN) of different components associated with snow (a), canopy (b), runoff (c), and external groundwater runoff (d). For more details about the components of GEOframe-Newage, the reader is referred to Formetta et al. (2014) and Bancheri et al. (2019).

which uses a sun-shade canopy model (de Pury and Farquhar, 1997) that solves the energy balance for both sun-lit and shaded vegetation, extending the one recently developed by Schymanski and Or (2017) to canopy level.

The idea behind creating a specific reservoir arrangement for the karst external catchment (AC) is that the fate of water in the soil is different in the surface catchment (SC) and the AC one. Therefore, a different arrangement of reservoirs has been included for CSA and MU (see Fig. 5(d)). Following the analysis done in Section 3, the AC water flowing into the CSA is conceived as a single reservoir with a travel-time parameter set to 30 days, while MU is modeled using a reservoir with a travel time equal to 167 days. In this study, the primary focus is on the temporal variation of the precipitation-recharge-discharge

behavior of the AC water flowing from CSA and Ussita rather than the spatial variability of the carbonate system's behavior. This allows us to specifically investigate the impact of single or multiple-year drought events on the basin storage, as discussed

in Section 5.2. So, a lumped modeling approach is more appropriate for providing more insights into the temporal system's response to drought conditions and its implications for basin storage. Furthermore, incorporating more spatial variability for the carbonate areas would result in an increased number of model parameters. This introduces additional uncertainty into the model. Given the limited availability of data for calibrating these parameters, using two separate lumped systems has been considered as an efficient strategy for the modeling. The results will also demonstrate that the temporal behavior of the AC water and its response to drought events could be investigated properly by this modeling approach.

## 4.2 Experiment setup, calibration, and validation

The model was calibrated with hourly river discharge, observed at the hydrometric stations of CSA, MU, and Visso, and cross-compared with EO products of evaporation and snow cover, as well as validated with river discharge observations. The comparison with EO data provides insights into the model's robustness in reproducing spatial patterns of ET and Snow across the study area.

The model was spun up at the CSA sub-basin in the period Jan 2005- Dec 2009, while the period from January 2010 to December 2015 was used to calibrate the model. Also for Visso, the warm-up period involved the first five years of data (Jan. 2005 - Dec. 2009), while the following years (Jan. 2010 to Dec. 2017) were used for calibration. Finally, the period from Jan. 2019 to December 2021 was applied for the calibration of model parameters for MU. The model is validated using only data from the Visso station between January 2019 and December 2021 because CSA data were not available in this period and MU data is quite short for calibrating the model. Note that the warm-up period was chosen based on the features of the sub-basins.

Based on the results of a sensitivity analysis (not shown), 18 parameters were chosen for the calibration process. The model calibration process was performed based on LUCA (Hay et al., 2006), using the available observed discharge at the mentioned stations. The calibration algorithm was used to maximize the Kling-Gupta Efficiency (KGE) (Gupta et al., 2009) value between the observed and simulated discharge time series.

In addition, different scores and hydrological signatures (Addor et al., 2017) were used to evaluate the optimized model robustness (Table 2), including mean daily discharge, high flow, low flow, flow duration curve slope, and low flow duration frequency. In particular, high flow and low flow represent the 95[th] and 5[th] percentile of the discharge, respectively. The low flow duration frequency signature is defined as the frequency of the days with discharge lower than 20% of the mean discharge value ($0.2\bar{Q}$). Since the Nera River discharge does not show significant variations, this signature is considered as the frequency of days with the discharges lower than the mean discharge value ($\bar{Q}$) instead of 20 % of the mean discharge ($0.2\bar{Q}$). On top of the classic goodness scores (i.e. NS and KGE) and mentioned signatures, the goodness of discharge simulations is evaluated in a more statistically elaborated way. This was done to understand the predictive uncertainty of the model better and thus the information brought by the model based on the observed river discharge. In particular, we extracted the most likely expected discharge and the range of probable discharge values by using an empirical-based conditional probability approach as described in the next section.

## 4.3 Assessing the reliability of the model with the Empirical Conditional Probability (ECP)

According to the literature, predictive uncertainty is defined as the probability of real values of any variable of interest (i.e., the predictand) conditional on all available information up to the present (Todini, 2008; Krzysztofowicz, 1999). This information is provided by any deterministic model (i.e., the predictor). According to this definition, we propose a method that identifies the conditional probability (i.e., the probability density of the predictand conditional on the model simulation, the predictor) based on a non-parametric parsimonious method to estimate the posterior distribution of a predictand. This approach overcomes the challenges of most predictive uncertainty-based statistical methods that have to deal with Gaussianity assumption (which can be far from reality in many cases) and problems such as extrapolation to extreme values.

The process of computing the Empirical Conditional Probability (ECP) involves the following steps:

- Combining the observed discharge and the corresponding simulated values into a single dataset.

- Grouping the dataset into $n$ classes (bins) according to the simulated discharge values. The quantile-based discretization method has been applied for binning data into different classes.

- Computing the Empirical Cumulative Distribution Function (ECDF) for each class $j$ using the formula:

$$ECDF_j(Q) = \frac{1}{m_j} \sum_{i=1}^{m_j} I_{X_i < Q} \tag{1}$$

Here, $ECDF$ represents the empirical cumulative distribution function of the $j$-th class, $m_j$ is the number of measures in the group, $X_i$ denotes the $i$-th measure in the group, and

$$I_{X_i < Q} = \begin{cases} 1 & \text{if } X_i < Q \\ 0 & \text{otherwise} \end{cases} \tag{2}$$

- Computing the features of empirical distribution function (i.e., mode, maximum, minimum, and mean of the discharge) for each class.

From the ECDF, we can derive the empirical distribution functions for different classes (e.g., the one shown in Fig. 6(b) and 6c), which are assigned to each time step. Fig. 6 shows an example of implementing this method at two time steps where the observed discharge values are equal to 3.6 and 4.51 $m^3/s$. In Fig. 6, the discharge corresponding to the highest probability and the observation discharge have been indicated with a green bar and a red line, respectively. The x-axis shows the range of probable discharge at each time step. For instance, as shown in Fig. 6 (b), the variation range of probable discharge values is between 3.25 $m^3/s$ to 5.1 $m^3/s$. The difference between the observed and the highest probable discharge value can be considered as an estimate of predictive model error. Fig. 6 (b) also shows that the observed discharge is well-matched with the highest probable discharge, so the model performance is considered reliable. On the other hand, for time step $t2$, as Fig. 6 (c) presents, the observed discharge does not match with the highest probable discharge value and the difference between observed discharge and highest probable discharge is a measure of model reliability.

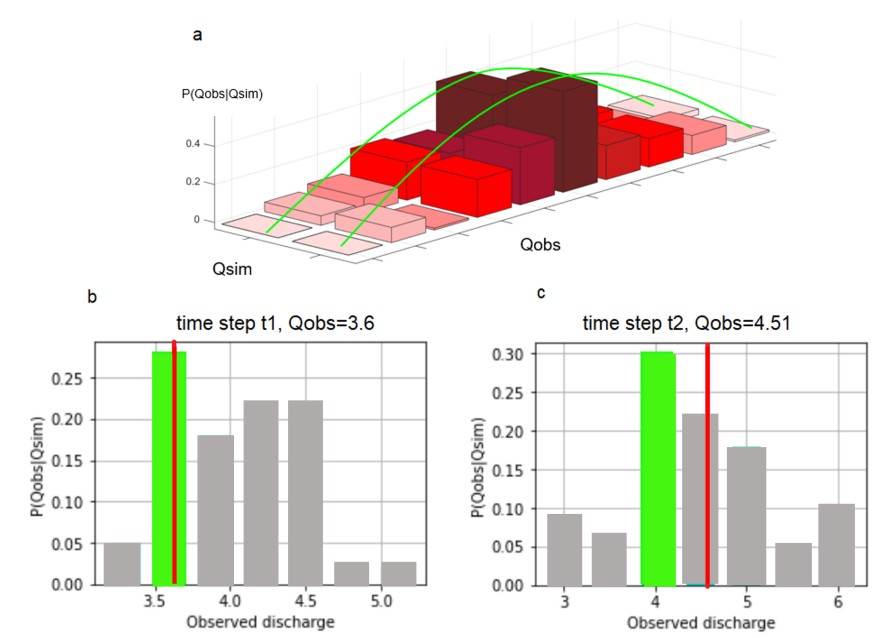

**Figure 6.** (a) Two empirical probability distribution functions (EPDF), represented as histograms, which are conditional on two different values of simulated discharge. (b) The examples of computing the predictive error at time step $t1$ with the observed discharge Q=3.61 m$^3$/s and (c) at time step $t2$ with the observed discharge Q=4.51 m$^3$/s in Visso basin. The green bar in each histogram represents the mode of the EPDF, while the observed discharge is represented by a red line. The difference between the observed and the most probable discharge is a proxy for the predictive model error.

The histograms obtained for different bins (e.g., Fig. 6 (b) and (c)) are dedicated to the time steps visualised as green dots and grey-shaded areas in Fig. 7, 8, and 9. The green dots and grey area illustrated in the figures provide an indicator of the reliability of the simulations, according to previous simulated and observed data. In particular, the disparity between the measured and the mode values (green dots) can be considered as a measure of this reliability. The complete estimation procedure is thoroughly documented in a specific Notebook, accessible in the supplementary material. It is important to highlight that the number of classes (bins) has been carefully chosen to ensure a meaningful histogram for each discharge class. Even for the shortest available dataset (at the MU station, which encompasses approximately 26,000 hourly data points), a reasonable number of samples are available for each discharge class.

## 5   Results

In this section, the evidence needed to answer the research questions (RQ) presented in the introduction is provided. In section 5.1, it is investigated whether the parsimonious modeling solution is efficient to model the water budget of complex systems (e.g., the Nera river basin with a huge external groundwater contribution). Moreover, different approaches to evaluate the

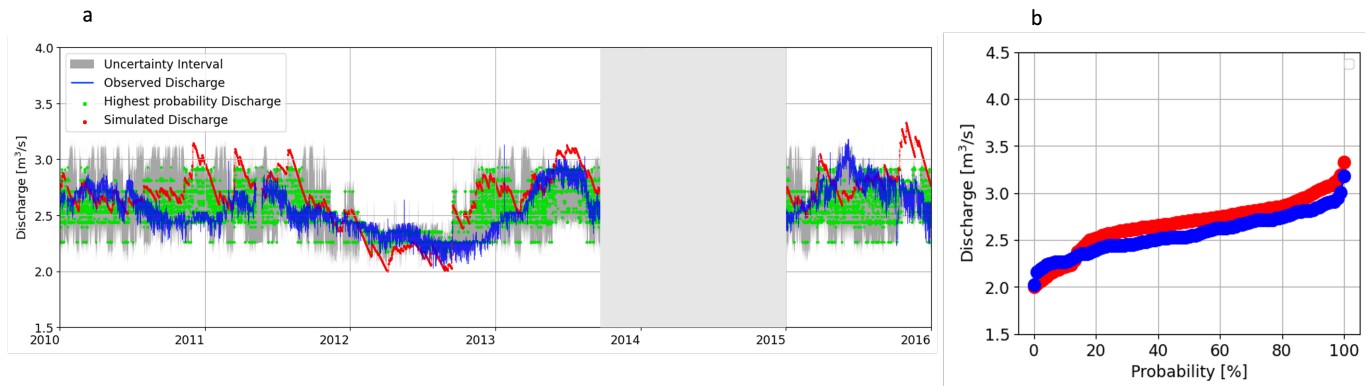

**Figure 7.** (a) Simulated discharge at the outlet of Castelsantangelo (CSA) during the calibration period. The part of discharge with low quality has been ignored from the analysis (light grey shaded area). The variation range of probable discharge values obtained by ECP shows in the grey-shaded area. The green dots show the value of the discharge with the highest probability in each time step. The closer to the observed discharge, the more reliable model performance is. The interval is not wide and it is an evidence of the capability of the model in the Castelsantangelo (CSA) discharge simulation. (b) The flow duration curve for the observed and modeled Castelsantangelo (CSA) discharge are in blue and red, respectively.

**Table 2.** Classic signatures applied to evaluate the performance of the model

| Type of Scores | Name | Description |
|---|---|---|
| General Scores | KGE (Gupta et al., 2009) | $KGE = 1 - \sqrt{(r-1)^2 + (\alpha-1)^2 + (\beta-1)^2}$ |
| | Correlation Coefficient | $R = \dfrac{\sum_{i=1}^{N}(S_i - \bar{M}_i)^2}{\sum_{i=1}^{N}(M_i - \bar{M}_i)^2}$ |
| High Flow | 95th percentile of streamflow | 95% flow quantile (high flow)(mm yr-1) |
| Low Flow | 5th percentile of streamflow | 5% flow quantile (lowflow) (mm yr -1) |
| LFD Frequency | Frequency of low flow days | |
| | (Westerberg and McMillan, 2015) | day yr-1 |
| Flow Duration Slope | Slope of flow duration | |
| | (Sawicz et al., 2011) | $SFD = \dfrac{ln(Q_{33\%}) - ln(Q_{66\%})}{0.66 - 0.33}$ |
| Mean Daily Discharge | - | mm day-1 |

model's reliability are investigated and eventually, a new statistical approach is proposed to understand the model's uncertainty

and reliability. Section 5.2 is concerned with the variability of different water budget components and the capability of the basin to sustain streamflow during single and multi-year drought episodes.

## 5.1 Model suitability to simulate river discharge (RQ1, RQ2, RQ3)

Fig. 7(a) shows the simulated hourly discharge at the outlet of CSA, considering the external reservoir in the model with its travel time parameter set to 30 days, as found in Section 3 and described in Section 4.1. During the calibration period, the KGE (see Table 2 ) and $R^2$ were obtained equal to 0.51 and 0.62, respectively. Furthermore, looking at Fig. 7(b) and the value of the slope of the flow duration curve for CSA in Table 3, it seems that there is not a considerable discrepancy between the high (95 [th] percentile) and low (5 [th] percentile) streamflow in CSA. Fig. 7(a) also shows predictive uncertainty results obtained with the ECP method. In this figure, red dots and blue line represent simulated and observed discharge, respectively. Due to the rapid frequency of oscillation, the discharge appears as clouds in the plot. Green dots represent the highest probable discharge based on ECP analysis. Higher differences between the green dots and blue line reveal the lower capacity of the model to predict discharge. Generally, the relatively small difference between the observed and the highest probable discharge values in each time step suggests the adequacy of the model to reproduce river discharge for this river section. In both Autumn 2011 and Autumn 2012, a notable disparity was observed between the simulated discharge (represented in red) and the actual measured discharge (depicted in blue). The pattern of the modal discharge (green dots) is also similar to the simulated one. This suggests that these two particular seasons deviated from the norm in comparison to other periods. Therefore, it would likely be beneficial to conduct separately more detailed studies for these anomalous seasons.

Further information derives from Table 3. It shows that for the majority of the days of the year, the discharge values are lower than the average river flow, indicating that the distribution of discharges is left-skewed. All the signatures in the Table are well reproduced, even though the calibration was not targeted at them directly. High flow statistics show a discrepancy of -5%, while low flow statistics bias is greater with a difference of around -10%. The duration curve of the simulated behavior is 30% steeper, meaning that the actual discharges are, on average, greater than the simulated ones, even though the latter have higher extremes. The discrepancy between simulated and observed mean discharge is, however, limited to less than 3%, which can arguably be considered below the heuristically expected uncertainty of the forecast. It should be noted that due to different periods of calibration at the three hydrometric stations, the average daily discharges cannot be compared to each other.

Fig. 8(a) shows the simulated discharge at the outlet of the basin (the Visso River station). The model simulation yields here $R^2$ and KGE values of 0.77 and 0.83 for the calibration period and 0.77 and 0.87 for the validation period, respectively, which is a sensible improvement with respect to CSA. Considering Table 3, the frequency of low flow (LFD) score for Visso, which is 205 days for the observed discharges, is similar to the observed value at CSA, demonstrating that low flows dominate the river even at the Visso outlet (except for the Nov. 2013 event). However, at Visso the model estimates more days with low flow than observed (211 vs. 205), while at CSA the opposite happens (180 vs. 208). In the same Table, a comparison of the slopes of the flow duration curves for Visso and CSA shows a lower oscillation of flow in CSA (see also Fig. 8(b)). Fig. 8(a) also shows how the model performs statistically with regard to historically observed discharge. The model accurately reproduces the jump

**Table 3.** Different signatures, after Addor et al. (2017) of simulated and observed flows during the calibration period at CSA, Visso, and MU hydrometric stations. It should be noted that the calibration periods are different for the three hydrometric stations and the average daily discharges cannot be compared with each other.

| Data | High Flow | Low Flow | LFD Frequency | Flow Duration Slope | Mean Daily Discharge |
|---|---|---|---|---|---|
| **CSA basin** | | | | | |
| Observation | 2.9 | 2.23 | 208 | 0.19 | 2.55 |
| Simulation | 2.78 | 2.03 | 180 | 0.25 | 2.48 |
| **Visso basin** | | | | | |
| Observation | 5.6 | 1.66 | 205 | 1.35 | 3.2 |
| Simulation | 5.19 | 1.90 | 211 | 1.03 | 3.15 |
| **MU basin** | | | | | |
| Observation | 3.6 | 1.59 | 198 | 0.85 | 2.46 |
| Simulation | 3.5 | 1.82 | 208 | 1.04 | 2.52 |

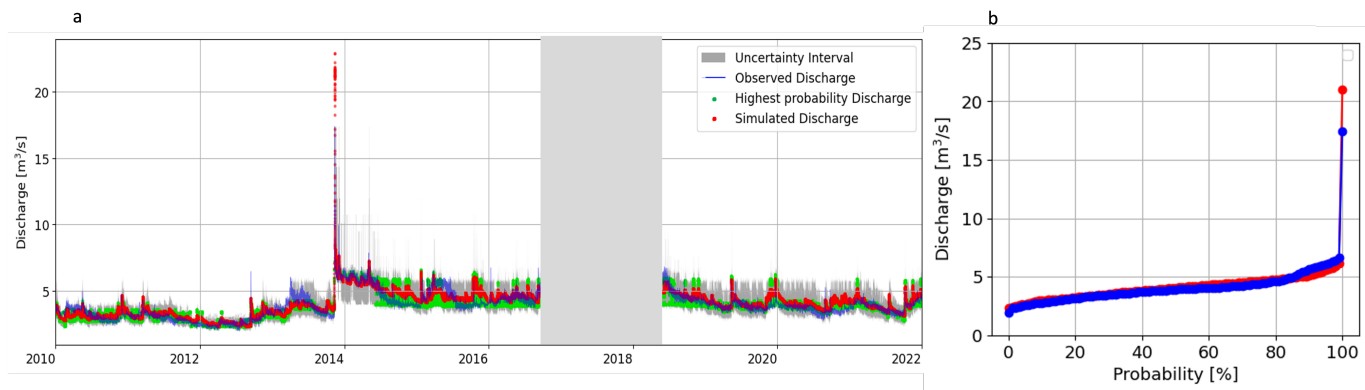

**Figure 8.** (a) Simulated discharge at the Visso outlet during the calibration (2008-2017) and validation periods (2019-2021), depicted in the same plot. The discharge affected by the earthquake has been ignored for analysis. The variation range of the probable observed discharge values in each time step is shown in the grey shaded area. The green dots show the value of the discharge with the highest probability in each time step. (b) The flow duration curve for observed and modeled Visso discharge are in blue and red, respectively.

in discharge observed in November 2014, although the average discharge is slightly overestimated. Overall, the green, red, and
295 blue lines exhibit close agreement.

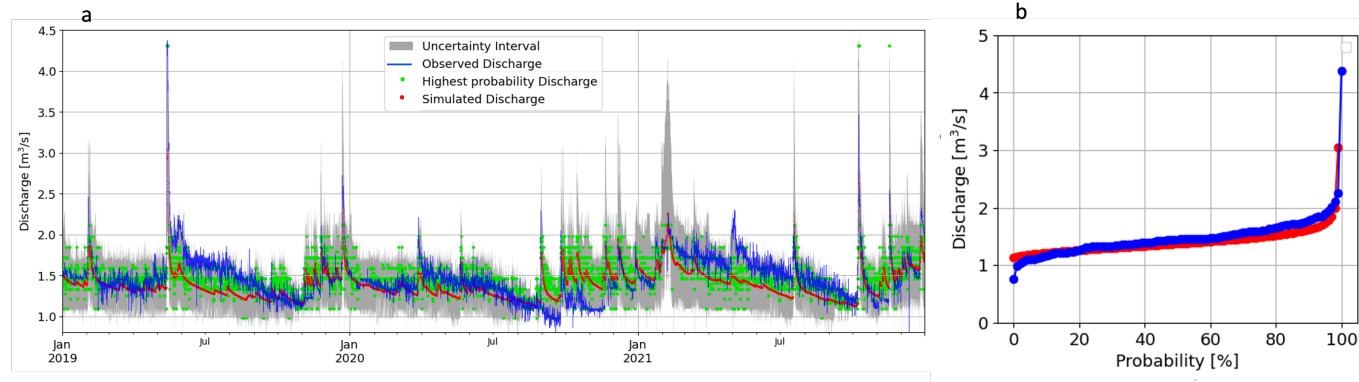

**Figure 9.** (a) Simulated discharges at the Madonna dell'Uccelletto (MU) outlet of the Ussita subbasin for the calibration period (2019-2021). The Uncertainty analysis results obtained by the ECP method for the simulated discharge are shown with the gray area (b) The flow duration curve for observed and modeled discharge are in blue and red, respectively.

Considering the travel time parameter of the external reservoir equal to 167 days (see Section 3 and 4.1), the $R^2$ and KGE values of the model for simulating MU river discharge during the calibration period were equal to 0.71 and 0.68, respectively. Fig. 9(a-b) also shows the simulated discharge at MU during the calibration period. Table 3 presents different scores related to the signatures, which show relatively good agreement. Although the long-term average value of discharge is almost the same at CSA and MU, the difference between low and high flows at MU is higher (Table 3). This confirms the dissimilar river regime behavior in these two parts of the Nera basin. Furthermore, the flow duration slope values in Table 3 demonstrate a higher variation of discharge at MU than CSA. Additionally, Fig. 9(a) shows the variation range of probable discharge at each time step -grey-shaded area- for MU which seems to be generally narrower compared to those of Visso, suggesting that the model has more uncertainty to give information about the river discharge variability at Visso station. It is interesting to note that this comes with a calibration score for simulating Visso discharge better than those of MU, providing evidence about the information that ECP can give on top of classical metrics.

### 5.1.1 Evapotranspiration

As was mentioned in the Methods section, ET was assessed by comparing GEOframe-Prospero ET (GET) and actual ET (MET) from MODIS product (Mu et al., 2013), based on the Penman-Monteith relation and various measured optical quantities. Figures 10(a1-a2) and (a3-a4) compare the GET and MET for two CSA sub-basins characterized by different elevations of 696 and 1,615 meters above sea level respectively. The two ET time series show good correlation, with correlation coefficients of 0.86 and 0.78 respectively. Furthermore, the figure shows that MET with much higher values than GET (i.e. it is less sensitive to stresses than Prospero).

Likewise, the comparison of GET and MET for two sub-basins located at 778 and 824 m a.s.l is shown in Fig. 10(b1-b2) and (b3-b4), with the correlation values of 0.76 and 0.77, respectively. In terms of bias, some positive systematic differences

were expected as the MET is partially based on large-scale forcings that embrace a much larger area at lower elevations than the study area. Comparing the plot of MET and GET reveals that higher ET values are more frequent in MET than in GET, resulting in a fatter tail for the MET than for the GET. The absolute difference between the two estimates can be used as an indicator of the error in these simulations.

The dynamics of ET is similar for both products but the peculiarity of the upper Nera catchment, located in a very mountainous basin, along with the uncertainty of the EO dataset makes the evaluation of the bias very uncertain. Overall we are more confident in the estimates provided by Prospero given that in a complex topography region like the one of the study area the radiation component of the ET can play an important role in evaporative fluxes and this is not explicitly considered in the MET product.

### 5.1.2   Snow

The current version of the GEOframe snow component provides the Snow Water Equivalent (SWE) and not snow depth. Thus, a comparison between the model results and the Sentinel-1 EO product was done only in terms of the spatial correlation between the snow cover obtained by the model and Sentinel-1. Fig. 11(a), and the associated box plot, shows the spatial correlation of Sentinel-1/GEOframe snow cover over time for CSA. To avoid uncertainties related to snow compaction (SWE vs snow depth), only the period from December to March is considered in the analysis, which corresponds to the snow accumulation period. The figure shows the 75th and 25th percentiles of correlation are in the range of 0.79 and 0.5 for the CSA basin.

Fig. 11(b) presents the spatial correlation between snow cover obtained by Sentinel-1 and GEOframe for the HRUs of the Nera River basin at Visso in the period of December to March varying quite widely. The 75th and 25th percentiles of correlation are in the range of 0.75 and 0.25, respectively. This agreement is relatively less for the MU sub-basin, as can be clearly seen in the same figure (Panel c), but we do not have sufficient elements to explain the reason for this. Overall, although the agreement of Sentinel-1 and in-situ snow depth has been investigated (Fig. 3S), further investigations on the ground are necessary for snow modeling.

### 5.2   Water budget analysis and the effect of groundwater discharge on the river regime (RQ4 and RQ5)

In this section, we present the interannual variability of the different water budget components of the catchment to understand the response of Mediterranean carbonate systems to climate variability. In particular, we focus on the ability of these basins to sustain streamflow during periods of significant precipitation deficit, such as the one experienced in the region in 2012 (Di Matteo et al., 2021b). For each delineated sub-catchment, the budget can be expressed as:

$$\frac{dS^\bullet}{dt} = P + Q^\bullet_{AC} - ET^\bullet - Q^\bullet \tag{3}$$

where $\bullet$ denotes quantities estimated by GEOframe, $dS/dt$ is the variation in water storage in the sub-catchment, $P$ is the total precipitation in the sub-catchment, $Q_{AC}$ is the external discharge supplied by fissured rocks, $ET$ is the evapotranspiration, and $Q$ is the discharge at the outlet. Note that the water budget is resolved over a hydrological year (from September of the

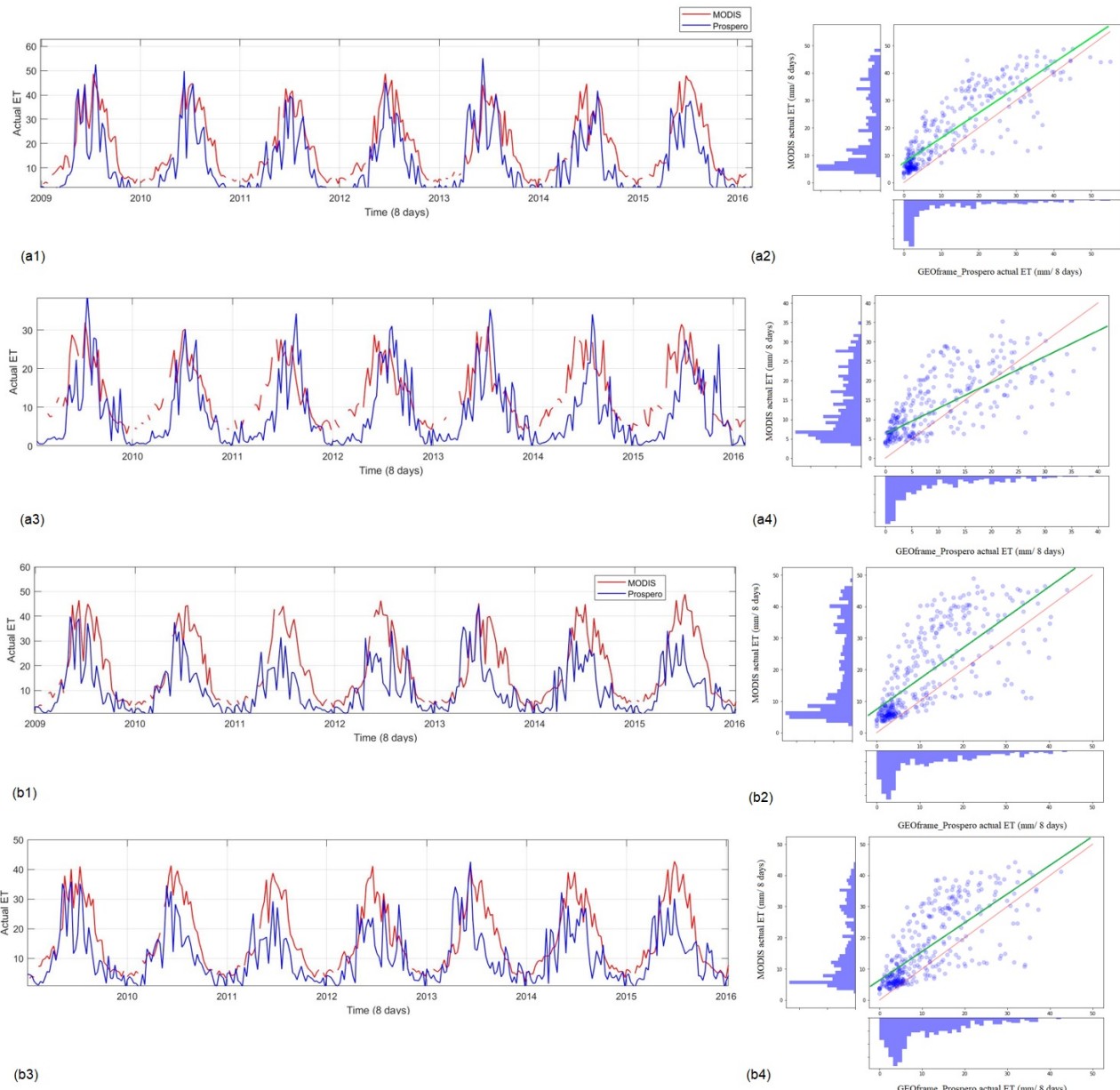

**Figure 10.** Comparison of actual evapotranspiration from MODIS and GEOframe-Prospero, and the associated scatter plots, for two Castelsantangelo sub-basins located at 696 m.a.s.l (a1 and a2) and 1615 m.a.s.l (a3 and a4), and two Nera sub-basins located at 778 m.a.s.l (b1 and b2) and 827 m.a.s.l (b3 and b4). The green and red lines in the scatter plots show the regression and 1:1 lines respectively. At higher elevations, a larger discrepancy between the MODIS and GEOframe-Prospero actual ET is observed.

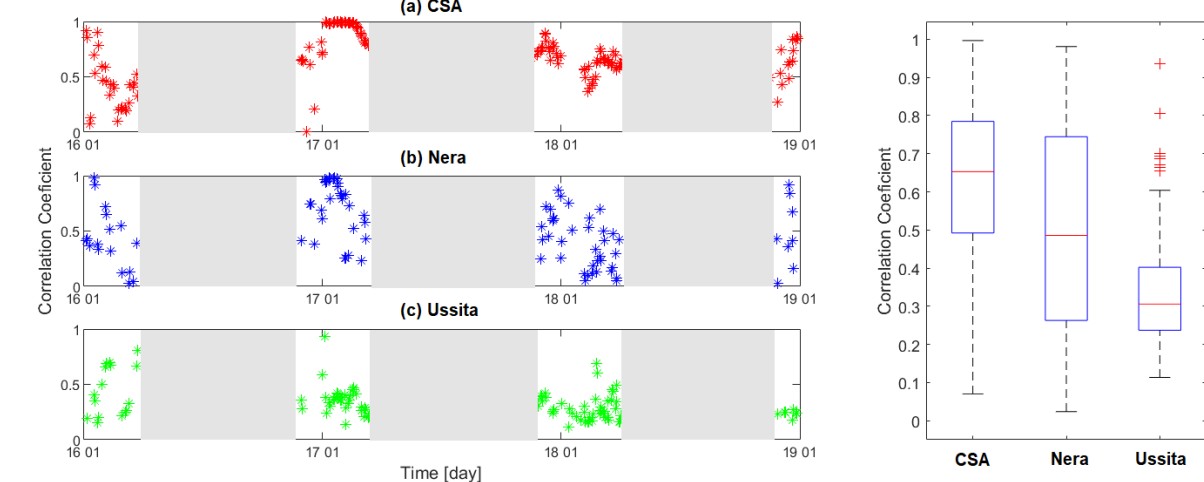

**Figure 11.** The spatial correlation of Sentinel-1 and GEOframe snow cover over time and the corresponding box plots for the (a) Castelsantangelo (CSA), (b) Nera, and (c) Ussita basins.

previous year to October of the current year). In this section, all the simulated values of the different components of the budget during the period 2010-2021 are considered, even the years impacted by the earthquake with different behavior.

Fig. 12 displays the different annual water budget components from 2010 to 2021, closed at CSA and Visso (outlet) stations. The carbonate areas and fissured rocks located upstream of CSA and MU are now called "external CSA" and "external Ussita", respectively. In the figure, for each year the left-hand bars are related to the input fluxes, including precipitation and the additional discharge supplied by the external CSA and external Ussita, the middle bars are associated with the output fluxes, which contain actual ET and river discharge at the outlet, and the right-hand bar is the variation of the basin storage. Fig. 12(a) (CSA) shows that the external CSA dominates the input fluxes and consequently the discharge at the outlet of CSA is significantly generated by this external area. Similarly, the external water from external Ussita and external CSA combined make a notable contribution to the input fluxes of the whole catchment (i.e., Visso station, Fig. 12 (b)), almost equaling the average precipitation falling within the hydrographic boundaries of the catchment.

Regarding the dynamics of the external CSA (panel a in Fig. 12), it can be seen that it is not constant, but rather it follows the interannual precipitation variability. For instance, during the dry year of 2012, the external CSA is significantly reduced with respect to the other years, while in 2014 the basins experienced the maximum annual precipitation for the observation period, resulting in the greatest increase of discharge from external CSA. Because of the low precipitation in 2012 CSA shows a slightly larger output flux than input flux, with a negative storage change, meaning that the basin was able to sustain river discharge during periods of significant precipitation deficit. For Visso (Fig. 12(b)) the storage differences remain positive for the period of interest, indicating a potentially infinite stored water accumulation over multiple years. Possible explanations are either that the basin feeds the groundwater system (not simulated by the model) between the CSA and Visso hydrometric stations (see also the observation discharge at CSA and Visso stations) or, considering the long-term memory of the basin, that ten years (2010-

2021) is a relatively short period for observing storage changes. While both assumptions remain unanswered, the complexity of the system makes the first assumption plausible, however further investigations are needed to provide compelling evidence of this. Moreover, the first assumption could also justify the lower runoff coefficient at Visso station.

Overall, ET plays a minor role at CSA but its contribution to Visso increases significantly. In a future drier and warmer Mediterranean, this could become a frequent situation, making this basin more vulnerable to drought episodes than others.

To better understand the catchment dynamics at Visso, in Fig. 13 we get rid of absolute values of the different water balance components and instead show the anomalies of precipitation, discharge, storage variation, and actual ET with respect to the average. Based on Fig. 13, it is clear that the pattern of anomalies for precipitation and storage are well in agreement. It can be seen that during dry years the storage anomalies are negative meaning that the storage is able to sustain river discharge in the catchment during these years. In particular, during single dry years like 2017 and 2019, the meteorological drought was significantly attenuated (precipitation deficit larger than river discharge deficit), while for subsequent dry years, like 2011 and 2012, the meteorological drought was slightly exacerbated in 2012 (precipitation deficit smaller than river discharge deficit). These results confirm the findings of Bruno et al. (2022), where a dataset of catchments in Italy indicated that carbonate basins are capable of attenuating meteorological droughts during single dry years. Also, the study of Alvarez-Garreton et al. (2021) shows that for basins characterized by large hydrological memory (i.e., large residence time of water within different stores e.g., snow, GW, soil moisture) multiple dry years can result in an exacerbation of the meteorological drought. This is explained by the recovery from drought for these basins, which is slower given the time that precipitation takes to replenish the depleted storages that sustained discharge during the previous drought year. This can be observed by comparing the anomalies of discharge in 2013 and 2018 in Fig. 13. Regarding evapotranspiration, a relatively constant pattern is observed, with a tendency to increase during 2012-2013, likely due to higher evaporative demand sustained by water stored in the unsaturated zone in weathered bedrock (i.e., rock moisture, Rempe and Dietrich (2018)).

# 6   Conclusions

Carbonate catchments supply a significant fraction of water for domestic water supply, energy production, agriculture, and industry, and they are strategic in dry climate areas like the Mediterranean region. Yet, the modeling of these complex systems is challenging because of the differences between the hydrogeological and hydrographic basin delineation. Regarding the malfunctionality of the conventional approach to delineate river basins, our findings demonstrated that it is still feasible to model complex carbonate rock catchments effectively by utilizing streamflow time series and additional information about the contributing area, specifically in case of lack of data. In particular, we leveraged the time series analysis to assist the hydrological modeling of the Upper Nera River basin – a complex fissured rock catchment located in the Apennines in central Italy. It is necessary to conduct a preliminary, yet straightforward, check on the water balance closure whenever doubts arise about the presence of external groundwater inputs in the river basin (see section 3). Specifically, for this study area the runoff coefficients in the upper part of the basin range from 4 to 5, indicating a substantial contribution of runoff in that particular area (Fig. 3 & 4). A time series analysis has been applied to determine the response times of the external catchments which

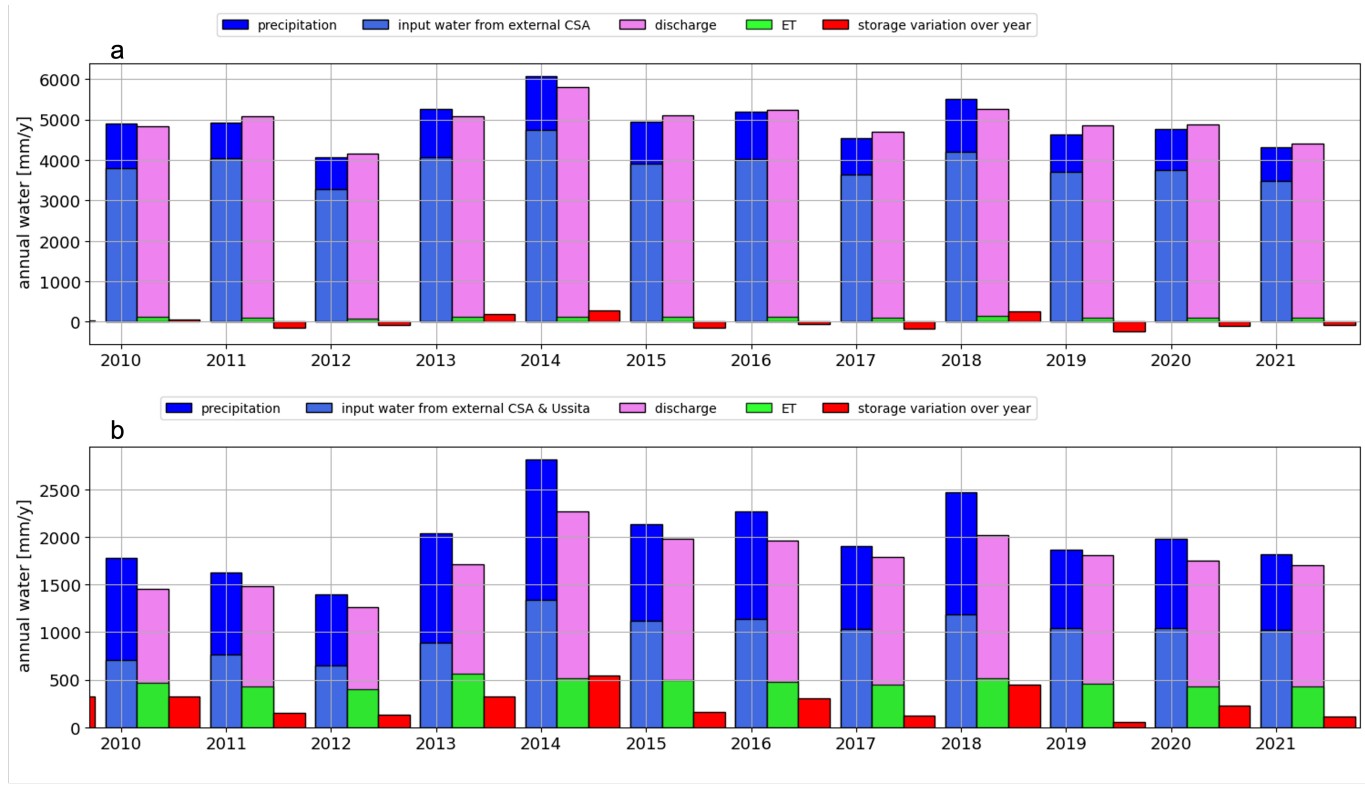

**Figure 12.** The annual water budget components during 2010-2021 over (a) Castelsantangelo (CSA), and (b) Visso (the whole catchment). The left-hand bars are related to the input fluxes i.e., precipitation and the additional discharge coming from the external Castelsantangelo (CSA) and external Ussita (purple bars), the middle bars are associated with the output fluxes i.e., actual ET and river discharge, and the right-hand bar is the variation of storage over each year.

are aligned with the estimates derived from field surveys incorporating tracer tests. Thereafter, by incorporating additional groundwater reservoirs into the modeling solution, with their average response time obtained through rainfall-discharge time series analysis, a significant improvement in the model's performance has been observed. This highlights the importance of considering the presence of groundwater reservoirs and their response times for accurately simulating the hydrological behaviour of the catchments. The configurable structure of GEOframe played a crucial role in the investigations, as it enabled

405     us to customize the modeling solutions according to the specific characteristics of the hydrological system in a straightforward manner. This flexibility allowed us to adapt the model to the unique features and complexities of the study area, ensuring a more accurate representation of the hydrological processes. Although difficulties still remain in river discharge estimation, we do not have sufficient elements to say whether these difficulties can be overcome with more complex and data-demanding modeling solutions. Overall, having more data with a longer period of overlapping records would be probably beneficial to

improve the simulation of such a complex basin behavior. Although one of the limitations of our study is the limited number of stations with overlapping records, employing a physically based (albeit lumped) modeling approach together with a robust

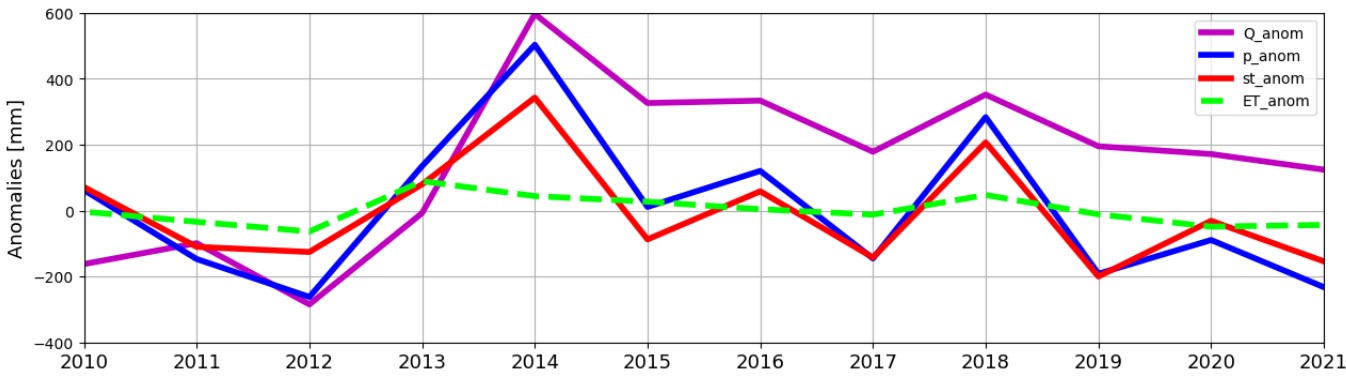

**Figure 13.** The anomalies pattern of annual precipitation, discharge, actual ET, and storage variation for the catchment during the water years of 2010-2021 at Visso station.

correlation analysis could mitigate the data shortage issue. The results also revealed that external groundwater discharge, primarily originating from fissured rocks, has a significant impact on the water budget of the basin, particularly in the upstream areas (CSA). Although this influence diminishes downstream, it remains a substantial component of the water budget, nearly equivalent to the average precipitation within the hydrographic boundaries of the basin.

To evaluate the reliability of simulated discharges, besides more traditional indicators, we employed a method based on the empirical probability of the observed discharge, conditioned on the simulated discharge. This methodology effectively assessed the model's performance. The relatively small variation range of probable discharge for CSA (the grey-shaded area) generally suggests the adequacy of the model to reproduce river discharge for this river section. The ECP analysis has provided compelling evidence that the discharge levels in Autumn 2011 and 2012 were anomalously low compared to the anticipated average. This finding strongly suggests that these seasons need further detailed investigation. Concurrently, these discrepancies underscore the uncertain inherent of parameter calibration. This remains true even for modelling procedures intended to be "physically based". Despite the better classic scores for simulating Visso discharge than those of MU, the variation range of probable discharge values at each time step for Visso seems to be generally wider than that of MU, suggesting that the model has more uncertainty to give information about the river discharge variability at Visso station.

Additionally, the model performance is cross-validated using earth observation ET and snow products. The results consistently showed a slight under-estimation of ET obtained by the GEOframe compared to the MODIS ET product. However, assuming MODIS-based ET as a reliable reference would imply lower discharges or reduced water accumulation in the groundwater system, which is inconsistent with the budget analysis of the CSA. While this discrepancy cannot be dismissed for the entire Nera basin, the CSA budget suggests opposition. Moreover, the model snow simulations are not in reliable agreement with EO snow information which warrants further investigation and analysis.

The water budget analysis of CSA shows that ET is not a significant component of the budget due to the substantial groundwater contribution from the carbonate area. The behaviour of soil moisture/groundwater storage at the CSA station oscillates

around zero, indicating a balance between recharge and discharge. The water budget analysis at the outlet of Visso shows a consistently positive accumulation of groundwater, which suggests the presence of a groundwater flow feeding by the river that is not adequately captured by the modelling solution. Therefore, further investigation is needed to better understand and incorporate these factors into the model to improve the representation of groundwater dynamics at the Visso closure.

We also examined the role of storage in sustaining river discharge during periods of significant precipitation deficit. The results revealed that during single dry years, such as 2017 or 2019, the anomaly of river discharge remained positive, indicating that groundwater storage played a crucial role in maintaining streamflow. However, during multi-year droughts, such as 2011-2012, slight drought conditions were observed.

Additionally, our research determined that periods of consecutive dry years led to a proportionally slower recovery of streamflow compared to single-year droughts as derived from the analysis of the discharge anomalies in 2013 and 2018. This discovery bears significant implications, especially in light of the Mediterranean region's trend towards increased aridity and warming, as underscored by preceding studies (Giorgi, 2006).

*Data availability.* The input data, simulations, simulated results, the Jupyter notebooks containing the analysis of the data, and the executable project are available on the paper OSF project at: https://osf.io/xtu4g/. The reader can download the OMS3 project, re-execute all the passages and check any part of what has been done in the paper. Everything can be used upon proper citation. Some works are required prior to any simulation with GEOframe. All of this is well documented through slides and videos on the GEOframe blog. The most recent material can be found at: https://geoframe.blogspot.com/2021/12/geoframe-winter-school-2022-gws2022.html. The Sentinel-1 snow depth retrievals are available online at https://ees.kuleuven.be/project/c-snow.

*Code availability.* We have made all the computer codes, data, and simulation schedules used in our study available for third-party inspection. This transparency allows for the reproducibility of our results and enables other researchers to validate and build upon our work. The availability of these resources promotes scientific collaboration and ensures the integrity of our findings. Interested parties can access and examine the materials to gain a deeper understanding of our research methodology and outcomes. The source code of the model components used in this paper are available at the GEOframe components Github site: https://github.com/geoframecomponents. The link https://github.com/giuliagiani/Tr_DMCA was used to retrieve the algorithm for the analysis by Giani et al. (2021).

*Author contributions.* All the Authors revised the paper and agreed on its contents. SA, CM, GF, SB and RR conceived the paper. SA, CM, GF and RR wrote the manuscript draft. GF wrote the Java codes specific to this paper. SA did most of the computations. SA and RR contributed the Python code for the analyses. AlT and DF performed the geological surveys, analyses and catchment identification in the field. SA, SB, SM and AT provided the EO data and did the quality check of the ground based data. CM, GF and RR provided support to the research with their projects.

*Competing interests.* We declare that no competing interests are present.

*Acknowledgements.* This paper has been partially supported by MIUR Project (PRIN 2020) "Unravelling interactions between WATER
and carbon cycles during drought and their impact on water resources and forest and grassland ecosySTEMs in the Mediterranean climate
(WATERSTEM) " (protocol code: 20202WF53Z ) and "WAFER" at CNR (Consiglio Nazionale delle Ricerche). The authors would like to
also thank the Monti Sibillini National Park, the Marche Region Authority: Ufficio Speciale Ricostruzione Marche and Centro Funzionale
di Protezione Civile Regione Marche (Graziano Candelaresi and Franscesca Sini). A special thank comes to Dr. Claudio Mariotti and Dr.
Tommaso Moramarco for the precious helps given in the understanding of the hydrological and geological characteristics of the area.

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
