# Peer review of "On understanding mountainous carbonate basins of the Mediterranean using parsimonious modeling solutions"

_EGUsphere, 2022_

## Author Comment (AC2)

**Reply to the Comments by Reviewer 1:**

**Rebuttal of the manuscript entitled "Resolving the water budget of a complex carbonate basin in Central Italy with parsimonious modelling solutions"**

We thank the Associate Editor and the reviewer for their valuable comments that, hopefully, will bring to a greatly enhanced manuscript. We have carefully read all the comments and provided point-to-point answers along with an indication of possible modifications to the manuscript. **Reviewers' comments** are in boldface. Underlined parts refer to changes in the revised manuscript.

**Reviewer's Comments:**

**I read this manuscript with great interest, knowing the complexity of modeling karst systems and the unique and important role they play in hydrology.**

**Reviewer's Comment 1: However, in the current reading of the discussion paper, the novelty of the work is not clear beyond the study area. The introduction states that the study demonstrated that "good results can still be obtained by using few experimental data and time series analysis"(L66) but the research experiment itself does not follow a systematic approach that shows how lesser and lesser data still results in similar results. Is the goal of the study to show that less data can still result in good modeling of a karst system or that it is not necessary to fully couple the surface-water and groundwater systems in a karst system (L65)? The study design does not seem to follow a systematic testing for either approach; rather it applies the GEOframe-Newage tools to the study area. It would be more compelling to compare these results to how the system could be modeling with other couplings or with more (or less) data.**

**Authors' Reply:**

We thank the reviewer for the valuable comment. The main novelty of this study is demonstrating that despite the lack of detailed groundwater (GW) information, yet, the modeling can be still conducted by extracting information from the observed hydrological time series within the basin and by relying on flexible modeling solutions.

Specifically, the novelties of our work are:

1. We were able to improve the appropriate goodness of fit indicators of simulated discharges of the karst system by using the obtained information from a new technique of correlation analysis between the rainfall and discharge time series.

2. We have proposed a new methodology based on the estimation of an Empirical Conditional Probability (EPC) which could provide information about the model reliability on top of the classical goodness of fit indicators.

The two previous achievements are methodological and can be applied to the other catchments, even for the non-carbonate ones.

However, the Nera catchment is an interesting study area by itself and is a good representative of the carbonate basins behavior. Therefore, our research experiment is extendable to other carbonate areas across the Mediterranean region. The accurate assessment, that the reviewer asks for, is the dream of any researcher but clearly is not achievable because of the missing appropriate groundwater data sets. In this regard, addressing the challenge of groundwater data scarcity and trying to provide reliable results (in terms of modelled and simulated discharges) is necessary and is the main scope of our study.

**Comment 1b: 'Reviewer asked the authors to specify clearly if the study scope is Showing that it is not necessary to fully couple the surface-water and groundwater systems in a karst system'**

**A:** Our contribution does not aim at demonstrating that fully-physical distributed models are not needed in these kinds of complex basins, rather, we are going to show that other effective solutions can be implemented especially when data availability is low. In other words, considering the fact that fully-physical modeling needs more expensive long-term groundwater data for the model calibration and run, we demonstrate here that a solution is still possible with a more parsimonious and fewer data demanding approach ( see Fig. 1). The Upper Nera River as the resource of multipurpose water supplies and as a representative of carbonate basins, with the problem of groundwater data shortage derived from climate change, is certainly of interest of many Mediterranean catchments that face similar challenges.

**Comment 1c: Reviewer asked the authors to specify clearly if the study scope is Showing that less data can still result in good modeling of a karst system.'**

**A:** We are not going to pursue the demonstration to which extent lesser and lesser data lead to similar results. During this study, we tried to show that in case of a lack of long-term groundwater data, still we are able to represent this complex interactions in a karst catchment in a reasonable way, if we know the response time of the groundwater system as deduced from the precipitation and discharge data and the extension of the external carbonate area. We actually we have also used the field work results of hydrogeologists to verify our hypothesis.

**Reviewer's Comment 2: The contribution is made more difficult to understand because the experiment organization in L70-79 appears to read more as results than hypotheses about what the study will test. The lack of clear hypotheses makes it difficult to understand the broader contribution of**

[Figure]

**Figure 1.** The proposed scheme in case of dealing with carbonate/karst systems without sufficient available groundwater data (the figure is for the reviewer and will not be added to the main text).

this work.

**Authors' Reply:**

We thank the reviewer for rising this up. We realized that the manuscript needs more work to define better our research hypothesis. Thus the text of those lines will be modified as follows:

"In this study, we are going to explore the following research questions:

1. Is it possible to model the complex carbonate catchment response to precipitation by relying only upon streamflow and precipitation time series? What kind of modelling solution is suitable to do that? And, is a parsimonious modeling solution suitable for that?

2. What is the impact of the external contributing area on streamflow in catchments characterized by fractured carbonate rocks behavior? And, to what extent does this contributing area impact the total streamflow from small headwater catchments to the main outlet?

3. What is the role of the storage for these types of catchments in sustaining streamflow during years of significant precipitation deficit?

We answer these questions on the Nera River basin, one of the main tributaries of the Tiber River (the second largest river in Italy) which contributes to almost 50% of its total discharge. Nera River is characterized by a significant portion of fissured and fractured carbonate rocks feeding the river discharge by releasing a large amount of groundwater into the riverbed by the streambed springs. Thus, this catchment is a good representative of the carbonate catchments to answer the three research questions.

Groundwater data shortage is a problem that is not unique to the Upper Nera River area, and the findings of this study could help inform water management and policy decisions in other carbonate basins as well. By providing a comprehensive analysis of the water cycle in this area, this study could also help identify

potential sources of water stress and may suggest strategies to mitigate them. Overall, this research has the potential to make a significant contribution to the understanding and management of water resources in carbonate basins and help ensure the long-term sustainability of the ecosystems of the basin."

**Reviewer's Comment 3**

**There are also quite a few qualitative statements that are not for the authors to decide about the quality of the modeling results. For example, on L234, the text states "these values are more than acceptable." It is not possible for the authors to make this assessment because they do not know what applications the readers may deem are "acceptable" - this is a qualitative statement based only on the authors' subjective assessment. The results should simply be reported and allow the reader to decide if these results are acceptable for their application or need. Another example is in L254, where the sentence reads, "It is apparent that the model is very good at reproducing the lowest discharges..." This should be changed to read something like, "The model is able to reproduce flows at the lowest discharges..." and then report or reference the accuracy at which the flows can be reproduced.**

**Authors' Reply:**

This point will be considered in the revised version of the manuscript as much as possible. We will polish the manuscript from these ambiguous expressions to make them more scientifically sound.

**Reviewer's Comment 4:**

**The results, interpretations, and discussion all relate specifically to the study area and there appears to be no further attempt to generalize or broaden the findings to the wider audience of HESS. There are also few stations used in the analysis, further limiting the interpretation of the results more widely. It would be helpful to frame these sections with a broader audience in mind beyond the study area.**

**Authors' Reply:**

We do agree in this context with the reviewer. In this regard, we will highlight the following points which make the manuscript more suitable for a wider range of journal readers:

1) In the modified Introduction, we will describe different approaches to take account of external groundwater flux contributions to a basin together with the advantages and drawbacks of each approach. Furthermore, we will describe the challenges of the lumped modeling approaches under the circumstance of data scarcity and then our strategies to address these challenges will be discussed. This will allow the paper to reach a broader audience.

2) We will consider the role of the storage for these types of catchments in sustaining streamflow during years of significant precipitation deficit. Nowadays, the European drought issue is at the core attention

of many researchers, and investigating this issue for the long-memory hydrological catchments (like the Nera River basin) would be interesting. Therefore, Section 5.2 will be totally modified.

3) The study area (Nera River basin) is a good representative of the carbonate basin behavior and makes our research experiment scalable to other carbonate areas across the Mediterranean region. Besides the importance of knowing the discharge of the Upper Nera River as the resource of multipurpose water supplies, this area as a representative of carbonate basins with the problem of groundwater data shortage could be of interest to those dealing with the carbonate basins with similar challenges.

**The Introduction will be rewritten as follows (subject to further revisions:**

"**Introduction**:

Carbonate/karst landscapes represent approximately 7-12 percent of the Earth's continental area and they provide a significant challenge for hydrologists (Hartmann et al. (2014)). Due to the capability of these landscapes in retaining water for a longer period (i.e., long-term hydrological memory catchments), the storage has an important role in the control of drought propagation and delayed hydrological recovery (Alvarez-Garreton et al. (2021)).

Generally, a carbonate/karst landscape forms when the percolated precipitation dissolves the bedrock in the subterranean carbonate/karst environments and creates extensive fissures, open fractures, conduits, and caves. This can result in a complex network of groundwater flowpaths occurring within the same or adjacent aquifers (Kiraly et al., 1995). Therefore, the hydrological simulations are generally more complex in these areas where external groundwater flow is significant.

For modelling carbonate/karst basins, different approaches have been used. One powerful solution is using distributed, process-based models (e.g., Rooji, 2020; Hartmann et al., 2014), which are based upon groundwater partial differential equation solvers. Yet, the main challenge of this kind of distributed model is that they require a large amount of hydrogeological data and extensive field analysis to set appropriate physical parameter values and correct boundary conditions. On top of that, a large computational power is needed to run these models (Li et al., 2022).

Alternative to these models is black-box models also based on machine learning in which all the details about the structure of the aquifer and the hydrodynamics parameters are not needed (e.g., Tapoglou et al. (2014) and Castilla-Rho et al. (2015)). Although the implementation of these models is easy, specifically for the areas where information is lacking, the main drawback is that their model parameters do not have a physical meaning and just are indirectly related to the characteristics of the carbonated system (Zhou et al., 2019). Furthermore, these models do not explicitly solve the water budget and thus it is not possible to have information about the dynamic of water budget components.

The other types of model are those called lumped models, which are based on a set of ordinary differential equations (ODEs) that conceptualize the whole carbonated system as a system of reservoirs without modeling explicitly its spatial variability (e.g., Hartmann et al., 2014; Rimmer and Hartmann, 2012; Butscher and Huggenberger, 2008; Tritz et al., 2011; Jukic and Denić-Jukić, 2009; Butscher and Huggenberger, 2008; Duboisl et al., 2020). However, even in this case, the definition of correct model parameters relies on calibration and inverse modeling using monitored discharge data (Hartmann et al., 2014). In

particular, there is a considerable amount of lumped modeling studies (e.g., Rimmer and Hartmann, 2012; Dubois et al., 2020) in which the fast and slow drainage from the carbonate system have been modeled by using different types of reservoirs. The parameters of these reservoirs have been obtained through calibration or, indirectly via tracers information (i.e., an artificial tracer is introduced into a sinkhole, and traces are then sought far away from it in the surrounding areas at different times (Hartmann et al., 2014; Zhang et al., 2021; Nanni et al., 2020)). Despite being useful, this technique is time-demanding and cannot be always implementable because of inaccessibility issues.

Techniques that use the correlation between precipitation and discharge can be valuable alternatives to understanding the behavior of carbonate systems, especially where there is a lack of field information about the water circulation in the basin. For instance, Fiorillo and Doglioni (2010) carried out a cross-correlation analysis to cope with the time that water needs to flow through the fissured aquifers. A useful method also has been borrowed from applied economics (Kristoufek, 2014, 2015) by Giani et al. (2021) to successfully estimate the basin response time of the hydrographs concerning the precipitation. However, so far, the later data analysis technique has not been applied to the complex carbonate systems to obtain the hydrological response to the precipitation.

In this study we explore the following three research questions (RQs):

1. Is it possible to model the complex carbonate catchment response to precipitation by relying only upon streamflow and precipitation time series? What kind of modelling solution is suitable to do that? And, is the parsimonious modeling solution suitable for that?

2. What is the impact of the external contributing area on streamflow in catchments characterized by fractured carbonate rocks behavior? And, to what extent does this contributing area impact the total streamflow from small headwater catchments to the main outlet?

3. What is the role of the storage for these types of catchments in sustaining streamflow during years of significant precipitation deficit?

We answer these questions on the Nera River basin, one of the main tributaries of the Tiber River (the second largest river in Italy) which contributes to almost 50% of its total discharge. Nera River is characterized by a significant portion of fissured and fractured carbonate rocks feeding the river discharge by releasing a large amount of groundwater into the river bed by the streambed springs. Thus, this catchment is a good representative of the carbonate catchments to answer the three RQs.

Groundwater data shortage is a problem that is not unique to the Upper Nera River area, and the findings of this study could help to inform water management and policy decisions in other carbonate basins as well. By providing a comprehensive analysis of the water cycle in this area, this study could also help identify potential sources of water stress and suggest strategies to mitigate them. Overall, this research has the potential to make a significant contribution to the understanding and management of water resources in carbonate basins and could help ensure the long-term sustainability of these vital ecosystems."

**Reviewer mentioned that "here are also few stations used in the analysis, further limiting the interpretation of the results more widely":**

**A:** In fact, this is the main challenge that we want to solve (i.e., implementing the model over regions with data scarcity issues).

We will add some comments on that in the discussion section of the revised manuscript to remark this.

**Reviewer's Comment 5**

**The conclusions make some interesting points, which actually do emphasize some of the potential novel aspects of the work but they are not emphasized in the manuscript elsewhere. For example, Conclusion #1 and the sentence on L356-357 discuss the insight that the classical approach for delineating basins is not appropriate and a preliminary check on the water balance is needed for karst system, especially if runoff coefficients are high. I am not sure of the novelty of this finding but this is a point that is noted in the title but then not mentioned again until the conclusions. The paper should be reframed with these contributions in mind. I will note again that I am not sure this will improve the novelty of the work but the conclusions are much more clearly stated as to the contribution of the work and it was unfortunate to wait until the end of the paper to understand the potential contributions of this work.**

**Authors' Reply:**

Thanks for the comment leading us to reframe the manuscript and to make everything cleaner.

**Other Issues:**

Minor comments will be all answered in the revised manuscript. Additional modifications that will be added to the next manuscript are as follows:

1. Based on the reviewer's comments, the Abstract will be changed to accomplish their suggestions.

2. We will improve Fig. 6, 7, and 9 in the original manuscript by highlighting the proposed evaluation method (based on empirical conditional probability) demonstrating that the general classical scores are not enough to evaluate the models.

3. Based on the reviewer's comments, the Conclusions will be modified also in the manuscript.

**References**

[revised manuscript text omitted]

---

## Author Comment (AC3)

**Reply to the Comments of Reviewer 2:**

**Rebuttal of the manuscript entitled "Resolving the water budget of a complex carbonate basin in Central Italy with parsimonious modelling solutions"**

We thank the Associate Editor and the reviewer for their valuable comments that certainly will improve our final manuscript. We have carefully read all the comments and provided the answers along with possible modifications. The reviewer made several suggestions about the wording. They were all accepted and are not reported here. **Reviewers' comments** are in boldface. Underlined parts refer to changes in the revised manuscript.

**Reviewer's Comment:**

**The manuscript from Shima et al. is dealing with a relevant topic, having potential interest for the readers of EGUsphere. The manuscript is well organized and containing useful information, sufficient data, good modeling efforts. Nevertheless, the goal of their research seems not well focused on novelties. In addition, the english language tremendously suffers for an misleading use of italian sentences which have been translated in english maintaining a classical italian structure.**

**Comment 1: The starting point is the non-correspondance of hydrographic basins and hydrogeological basins. This finding is a very basic one, everyone knows the difference and it is absolutely non a novelty for the scientific community. In their text, the Authors are presenting this issue as a novelty, instead of presenting the problem in the introduction chapter, using the relevand and abundant literature on this topic. So, the novelty of their manuscript has to be searched in the methods they applied to solve the problem. This is in my opinion the logical approach and I suggest them to completely rewrite the introduction focusing on the problem they want to analyse (how to take into account the overflow in river discharge due to external groundwater flow feeding your basin). By this way, they can easily highlight their findings, mainly related to the useful modeling and methodology they performed during the study.**

**Authors' Reply:**

We thank the reviewer for the valuable comment. Our study is dealing with the simulation of the complex hydrological behavior of basins where there is not correspondence between hydrological and hydrogeological contributing areas. The existence of these kinds of basins is not certainly a novelty, but what we have proposed with this paper is how to deal with them in cases of data scarcity. Besides, we introduce a couple of new tools and the use of data from tracers collected by some of the Authors. From the reviewer's

comments, we realized that this message was not clear in the original version of the manuscript, and in the revised manuscript we will change the Introduction, Results, and Conclusions sections.

In particular, the introduction will be re-organized as follows:

1. We will explain the general characteristics of the karst system.

2. We will mention different approaches of taking account of external groundwater flux contributions to a basin together with the advantages and drawbacks of each approach. Furthermore, we will describe the challenges of the lumped modeling approaches versus the fully distributed ones. This will allow the paper to be interesting for a broader audience.

3. We will clarify the study objective in three main research questions (RQs) which will be answered in different sections of the manuscript.

Three research questions are as follows:

1. Is it possible to model the complex carbonate catchment response to precipitation by relying only upon streamflow and precipitation time series? What kind of modelling solution is suitable to do that? And, is the parsimonious modeling solution suitable for that?

2. What is the impact of the external contributing area on streamflow in catchments characterized by fractured carbonate rocks behaviour? And, to what extent does this contributing area impact the total streamflow from small headwater catchments to the main outlet?

3. What is the role of the storage for these types of catchments in sustaining streamflow during years of significant precipitation deficit?

A proposal for the new Introduction can be found in the answer to reviewer's #1 comments.

**Reviewer's Comment 2**
**The second concern is related to the English language. Too many sentences are too long, with secondary sentences included. The uses of commas is limited and this approach cannot be approved by international readers. Please rewrite the entire document using shorter and clear sentences: one concept, one phrase. I strongly suggest the support of a mothertongue for providing a successful review.**

**Authors' Reply:**

The manuscript will be reviewed by a native English speaker.
Herein below we will try to answer the "Detailed comments". The other ones related to the typos and incorrect use of English words were all accepted and are not reported here.

**Detailed Comments**

1. **line 92: what do you intend with "linear" springs? Perhaps "streambed" springs?**

   **Authors:** Yes, thanks for the suggestion. We will replace the term.

2. **120: My?**
   **Authors:** This is a typo. It should be MU which is the acronym for Madonna dell'Uccelletto. It will be modified.

3. **line 160: you are in a karst domain, so a response in 3 days would be due to karst circuits. Please evaluate this possibility and if you exclude this possibility please explain why**

   **Authors:** According to Petitta et al. (2022), the continental deposits preserve this carbonate aquifer from the direct dissolution processes limiting the mature karst development in the saturated zones. Additionally, they demonstrated that the fast flow contributes to only a minor percentage of the discharge in this area, and the groundwater circulation is mainly driven by fractures and fissures. The three days delay is already well trated by a groundwater linear reservoir. That is why we have considered just the 30 days as the response time od the karst catchment (as highlighted also in Nanni et al. (2020)). However, we will modify the manuscript to clarify that the study area is not considered as a fully karst system.

4. **caption of figure 3: " is still high" is qualitative evaluation, please specify the number (it seems that in this case is lower than 1, so why you think is high?)**

   **Authors:** We will modify the caption of Fig. 3: "(a1) Cumulative observed discharge at CSA versus cumulative precipitation recorded at the closest station to CSA; (a2) Coefficient time series computed by dividing the discharge at CSA by the precipitation time series recorded at different stations. (b1) Cumulative observed discharge at Visso versus cumulative precipitation related to a station close to Visso; (b2) Coefficient time series computed by dividing the discharge at Visso by the precipitation observed at several stations. (c1) Cumulative observed discharge at Ussita versus cumulative precipitation related to a station close to Ussita; (c2) Coefficient time series computed by dividing the discharge at Ussita by the precipitation observed at several stations. The 1:1 line is shown in green. For CSA and Ussita the runoff coefficient is about 4 and 1.5, respectively, and this value is around 1 at the outlet of the basin (Visso)"

5. **line 205: using the period 2017-2018, do you not have problems with the reaction to the earth-quake? I read some papers indicating a long reaction in discharge in this zone**

   **Authors:** We have not considered the discharge records affected by the seismic sequences during 2016-2017 (see Fig. 7 and 9 in the manuscript). In the revised manuscript, we will modify the period of calibration for MU to 2018-2021. The model will be then cross-validated at the outlet of the basin (Visso) during 2019-2021. In this regard, Fig. 1 in this rebuttal is the modified version of Fig. 9 in the manuscript and it will be replaced.

[Figure]

**Figure 1. This figure will be Fig. 9 in the manuscript:** (a) Simulated discharges at the Ussita outlet for the calibration period (2018-2021). The dark and light blue-shaded parts are related to the carbonated and total discharge drained at Ussita station, respectively. (b) The flow duration curve for observed and modeled discharge are in red and blue, respectively. (c) The Uncertainty analysis results for the simulated discharge obtained by the ECP method.

115

6. **line 297: this is the real core of your manuscript and this has to be highlighted both in the discussion and in the conclusion!**

**Authors:** Thanks for the useful comment. The content of this section is related to the water budget analysis and also the figures will be reorganized. Additionally, we will highlight the important role of storage in supporting the river discharge in the years with precipitation deficit. Furthermore, the recovery from dry years in these kinds of basins with long hydrological memory will be discussed. Section 5.2 will be totally modified.

7. **line 321: where is Pescara spring? Out of your study area? So why you includes this spring in your comments here? I suggest to cancel this reference**

**Authors:** We will remove it from the text to avoid confusion.

8. **line 324: I did not find a "lack of clear recharge signal" in the reference you cited here. I suggest to cancel this part, not necessary and not included in your study area**
   **Authors:** We will remove it from the text accordingly.

9. **line 326: the sentence is not clear, please rephrase the concept. I know that aquifer recharge is EVER going to springs/river, producing discharge (not runoff)**
   **Authors:** We will remove the sentence to avoid any ambiguity.

10. **line 328: you have not discussed the role of Karst, so I suggest to not include karst in the conclusion**
    **Authors:** Based on point 3 the basin cannot be considered fully karst so the text will be modified accordingly.

11. **line 332: if you have karst, please discuss in the text, not in the conclusion**
    **Authors:** We will re-organize the text to avoid ambiguity (see points 3 and 10). Thanks for mentioning this point.

12. **line 357: your findings are not based on isotopes neither in tracer tests, so why you added in the conclusion?**

    **Authors:** The reviewer is right. We will modify the Conclusions accordingly.

**Authors:**

ADDITIONAL modifications that will be added to the manuscript are as follows:

– Based on the reviewer's comments, the abstract will be changed.

– The Introduction will be rewritten.

– We will improve Fig. 7, 8, and 9 in the original manuscript by highlighting the proposed evaluation method (based on empirical conditional probability) demonstrating that the general classical scores are not enough to evaluate the models.

**References**

Nanni, T., Vivalda, P. M., Palpacelli, S., Marcellini, M., and Tazioli, A.: Groundwater circulation and earthquake-related changes in hydro-geological karst environments: a case study of the Sibillini Mountains (central Italy) involving artificial tracers., Hydrogeology Journal, 28, 2409–2428, https://doi.org/https://doi.org/10.1007/s10040-020-02207-w, 2020.

Petitta, M., Banzato, F., Lorenzi, V., Matani, E., and Sbarbati, C.: Determining recharge distribution in fractured carbonate aquifers in central Italy using environmental isotopes: snowpack cover as an indicator for future availability of groundwater resources., Hydrogeology Journal, 10, 1619—-1636, 2022.

165

---

## Author Response (AR1)

**Reply to the Comments by Reviewer 1:**

**Rebuttal of the manuscript entitled "On understanding mountainous carbonate basins of the Mediterranean using parsimonious modeling solutions"**

We thank the Associate Editor and the reviewer for their remarkable and constructive comments. We have carefully read all the comments and provided point-by-point answers along with an indication of possible modifications to the manuscript. **Reviewers' comments** are in boldface.

The colour of the text in the revised manuscript is as follows:

1. blue is used to specify the changes related to reviewer 1's comments.

2. cyan is used to specify the changes related to reviewer 2's comments.

3. red is showing the changes which commonly refer to both reviewers' comments.

**Reviewer's Comments:**

**I read this manuscript with great interest, knowing the complexity of modeling karst systems and the unique and important role they play in hydrology.**

**Reviewer's Comment 1: However, in the current reading of the discussion paper, the novelty of the work is not clear beyond the study area. The introduction states that the study demonstrated that "good results can still be obtained by using few experimental data and time series analysis"(L66) but the research experiment itself does not follow a systematic approach that shows how lesser and lesser data still results in similar results. Is the goal of the study to show that less data can still result in good modeling of a karst system or that it is not necessary to fully couple the surface-water and groundwater systems in a karst system (L65)? The study design does not seem to follow a systematic testing for either approach; rather it applies the GEOframe-Newage tools to the study area. It would be more compelling to compare these results to how the system could be modeling with other couplings or with more (or less) data.**

**Authors' Reply:**
We thank the reviewer for the valuable comment that prompted us to clarify our primary objectives. The novelty of our study lies in developing an approach to model carbonate/karst catchments -under the circumstance of groundwater data shortage- taking advantage of flexible modeling approaches and rainfall-discharge time series analysis. Considering these kinds of catchments becomes more important since of their vulnerability and effectiveness to drought and climate change conditions.

The innovations in our study can be summarized as follows:

1. We have introduced a novel approach by taking advantage of a correlation analysis technique to extract groundwater (GW) hydrological response to precipitation. This technique has not been previously applied to carbonate systems for determining GW hydrological response. By incorporating this information, we have enhanced the reliability of the hydrological modeling of a complex carbonate rock catchment.

2. In addition to traditional goodness-of-fit indicators, we have proposed a new methodology called Empirical Conditional Probability (EPC). This approach provides valuable insights into the reliability of the model, offering a comprehensive assessment of its performance.

We tried to clarify the novelties and corresponding highlighted findings in the revised Introduction and in the Conclusions section.

The proposed methodological tools are not limited to the specific catchment studied in this research. They can be applied to other catchments, including non-carbonate ones. However, the Nera catchment itself is an intriguing study area and serves as a representative example of the behaviour of carbonate basins. Consequently, our experimental findings can be extended to other carbonate regions across the Mediterranean, providing valuable insights into their hydrological dynamics. We address these points in **Lines 57-62, Lines 201-232, Lines 370-406** of the revised text.

**Comment 1b: 'Reviewer asked the authors to specify clearly if the study scope is Showing that it is not necessary to fully couple the surface-water and groundwater systems in a karst system'**

**A:** Our contribution does not seek to prove that fully-physical distributed models are unnecessary in complex basins. Instead, we aim to demonstrate that alternative effective solutions can be implemented, particularly when data availability is limited. In essence, we acknowledge that fully-physical modeling requires extensive and costly long-term groundwater data for calibration and execution. However, our research illustrates that a viable solution can still be achieved through a more parsimonious and less data-intensive approach (refer to Fig. 1). To clarify this **Lines 159-162 and 388-390 have been added to the main text.**

**Comment 1c: Reviewer asked the authors to specify clearly if the study scope is Showing that less data can still result in good modeling of a karst system.'**

**A:** Understanding to what extent reduced data can still yield comparable results is not the scope of our study, since to satisfy this target an abundance of groundwater data is needed that is typically unavailable. Instead, our objective was to demonstrate that even in the absence of long-term groundwater data, it is still possible to reasonably represent the intricate interactions within a carbonate catchment by utilizing precipitation and discharge data, along with an estimation of the external carbonate area. To validate our hypothesis, we also incorporated fieldwork findings from hydrogeologists. **Lines 1-8, 58-60 (research questions 1 and 2), and 372-377** refer to this comment in the revised version of the manuscript.

rev.png

[Figure]

**Figure 1.** The proposed scheme in case of dealing with carbonate/karst systems without sufficient available groundwater data (the figure is for the reviewer and is not be added to the main text).

**Reviewer's Comment 2: The contribution is made more difficult to understand because the experiment organization in L70-79 appears to read more as results than hypotheses about what the study will test. The lack of clear hypotheses makes it difficult to understand the broader contribution of this work.**

**Authors' Reply:**

We thank the reviewer for raising this. We realized that the manuscript needs more effort to define better our research hypothesis. Thus we clearly mentioned the research questions at the end of the introduction, as follows (**see Lines 56-73 in the revised text**):

" This study aims to address the following five research questions (RQs):

1. Can the complex response of carbonate catchments to precipitation be modeled with HDSys relying only upon streamflow and precipitation time series, aided by cross-correlation analysis?

2. What type of modeling solution is suitable for this task, and is a parsimonious modeling approach appropriate?

3. Are the classic goodness-of-fit scores enough to evaluate the reliability of the models?

4. What is the impact of external contributing areas on streamflow in catchments with fractured carbonate rocks? To what extent does this contributing area affect the total streamflow from small headwater catchments to the main outlet?

5. What is the role of storage in sustaining streamflow during years with significant precipitation deficit in these types of catchment?

We have examined the water budget of the Nera River basin, which is a significant tributary of the Tiber River, the second-largest river in Italy. The Nera River basin contributes nearly 50% of the total discharge of the Tiber River and is characterized by a significant portion of fissured and fractured carbonate rocks feeding the river discharge by releasing a large amount of groundwater into the river bed from streambed springs. Thus, this catchment is a good representative of the carbonate catchments for answering the RQs."

**Reviewer's Comment 3**

**There are also quite a few qualitative statements that are not for the authors to decide about the quality of the modeling results. For example, on L234, the text states "these values are more than acceptable." It is not possible for the authors to make this assessment because they do not know what applications the readers may deem are "acceptable" - this is a qualitative statement based only on the authors' subjective assessment. The results should simply be reported and allow the reader to decide if these results are acceptable for their application or need. Another example is in L254, where the sentence reads, "It is apparent that the model is very good at reproducing the lowest discharges..." This should be changed to read something like, "The model is able to reproduce flows at the lowest discharges..." and then report or reference the accuracy at which the flows can be reproduced.**

**Authors' Reply:**

In the revised version of the manuscript, we have made significant improvements to address the concerns raised about ambiguous expressions and enhance the rigor of our writing. We have carefully reviewed and polished the manuscript to ensure that it accurately conveys our findings and methodology.

**Reviewer's Comment 4:**

**The results, interpretations, and discussion all relate specifically to the study area and there appears to be no further attempt to generalize or broaden the findings to the wider audience of HESS. There are also few stations used in the analysis, further limiting the interpretation of the results more widely. It would be helpful to frame these sections with a broader audience in mind beyond the study area.**

**Authors' Reply:**

We do agree in this context with the reviewer. In this regard, we highlighted the following points which make the manuscript more suitable for a wider range of journal readers:

1) In the modified Introduction, we describe different approaches to take account of external groundwater contributions to a basin together with the advantages and drawbacks of each approach. Furthermore, we explain the challenges of the lumped modeling approaches under the circumstance of data scarcity and then our strategies to address these challenges have been discussed. This allows the paper to reach a broader audience (**please see the Introduction of the revised manuscript**).

2) We consider the role of the storage for these types of catchments in sustaining streamflow during years of significant precipitation deficit. Nowadays, the European drought issue is at the core attention of many researchers, and investigating this issue for the long-memory hydrological catchments (like the Nera River basin) is interesting for the readers or the journal. Therefore, considering the drought point of view, **Section 5.2 (Line 320-368)** has been totally modified.

3) The study area (Nera River basin) is a good representative of the carbonate basin behavior and makes our research experiment scalable to other carbonate areas across the Mediterranean region. Besides the importance of knowing the discharge of the Upper Nera River as the resource of multipurpose water supplies, this area as a representative of carbonate basins with the problem of groundwater data shortage could be of interest to those dealing with the carbonate basins with similar challenges (**Lines 66-73 have been added to the text**).

**The Introduction has been rewritten** as follows:

"**Introduction**:

Carbonate/karst landscapes represent approximately 7-12 percent of the Earth's continental area and they provide a significant challenge for hydrologists (Hartmann et al. (2014)). Due to the capability of these landscapes to retain water for a longer period (i.e., long-term hydrological memory catchments), their storage plays an important role in the control of drought propagation and delayed hydrological recovery (Alvarez-Garreton et al. (2021)).

Generally, a carbonate/karst landscape forms when the percolated precipitation dissolves the subterranean carbonate bedrock and creates extensive fissures, open fractures, conduits, and caves. This can result in a complex network of groundwater flowpaths occurring within the same or adjacent aquifers (Kiraly et al., 1995). To model these types of systems one powerful solution is to use distributed, process-based models (PB) (e.g., Rooji, 2020; Hartmann et al., 2014), which are based on solvers for groundwater partial differential equation. Yet, the main challenge of this kind of distributed model is that they require a large amount of hydrogeological data and extensive field analysis to set appropriate physical parameter values and correct boundary conditions. On top of that, large computational power is needed to run these models (Li et al., 2022).

An alternative to PB is black-box models based on machine learning (MLM) in which all the details about the structure of the aquifer and the hydrodynamics parameters are not needed (e.g., Tapoglou et al. (2014) and Castilla-Rho et al. (2015)). Although the implementation of MLM is easy, their model parameters do not have a physical meaning and are only indirectly related to the characteristics of the carbonate system (Zhou et al., 2019). Furthermore, MLM does not explicitly solve the water budget and thus it is not possible to have information about the dynamics of all water budget components.

Hydrological Dynamical Systems (lumped models, HDSys) represent another type of model, based on a set of ordinary differential equations (ODEs) that conceptualize the entire carbonate system as a series of reservoirs (e.g., Bancheri et al., 2019; Hartmann et al., 2014; Rimmer and Hartmann, 2012; Butscher and Huggenberger, 2008; Tritz et al., 2011; Jukic and Denić-Jukić, 2009; Duboisl et al., 2020). Instead

of explicitly considering spatial variables, HDSys specify the interconnection of fluxes between different reservoirs, which leads to reducing the computational complexity. However, HDSys still require the definition of model parameters, which typically rely on calibration and inverse modeling using monitored discharge data or other relevant data sources (Hartmann et al., 2014). Several studies have also explored modeling the fast and slow drainage from carbonate systems using tracer information (e.g., Rimmer and Hartmann, 2012; Dubois et al., 2020). This approach involves introducing an artificial tracer into a sinkhole and then tracking the tracer's movement in the surrounding areas at different times (Hartmann et al., 2014; Zhang et al., 2021; Nanni et al., 2020). While this technique can be useful, it is time-consuming and may not always be feasible due to accessibility issues.

These HDSys can be conjugated by techniques that rely on the correlation between precipitation and discharge can provide valuable insights about the behavior of carbonate systems, particularly in situations where field information about water circulation is limited. It could also provide useful information in the situation of missing tracer test analysis. For example, Fiorillo and Doglioni (2010) used cross-correlation analysis to estimate the time that water requires to flow through fissured aquifers. Another useful method, borrowed from applied economics (Kristoufek, 2014, 2015), was employed by Giani et al. (2021) to estimate the basin response time of hydrographs to precipitation, with successful results. However, according to the authors' best knowledge, to date, this data analysis technique has not been applied to complex carbonate systems to determine their hydrological response to precipitation.

This study aims to address the following five research questions (RQs):

1. Can the complex response of carbonate catchments to precipitation be modeled with HDSys relying only upon streamflow and precipitation time series, aided by cross-correlation analysis?

2. What type of modeling solution is suitable for this task, and is a parsimonious modeling approach appropriate?

3. Are the classic goodness of fit scores enough to evaluate the reliability of the models?

4. What is the impact of external contributing areas on streamflow in catchments with fractured carbonate rocks? To what extent does this contributing area affect the total streamflow from small headwater catchments to the main outlet?

5. What is the role of storage in sustaining streamflow during the years with significant precipitation deficit in these types of catchments?

We have examined the water budget of the Nera River basin, which is a significant tributary of the Tiber River, the second-largest river in Italy. The Nera River basin contributes nearly 50% of the total discharge of the Tiber River and is characterized by a significant portion of fissured and fractured carbonate rocks feeding the river discharge by releasing a large amount of groundwater into the river bed from streambed springs. Thus, this catchment is a good representative of the carbonate catchments for answering the RQs. Additionally, groundwater data shortage is a problem that is not unique to the Upper Nera River area, and the findings of this study could help inform water management and policy decisions in other carbonate basins as well. By providing a comprehensive analysis of the water cycle in this area, this study could also help identify potential sources of water stress and suggest strategies to mitigate them. "

**Reviewer mentioned that "here are also few stations used in the analysis, further limiting the interpretation of the results more widely":**

**A:** In fact, this is the main challenge that we want to solve (i.e., implementing the model over regions with data scarcity issues).

We added some comments on that in the revised manuscript to remark on this (**like Lines 1-4 and 70-72.**).

**Reviewer's Comment 5**

**The conclusions make some interesting points, which actually do emphasize some of the potential novel aspects of the work but they are not emphasized in the manuscript elsewhere. For example, Conclusion #1 and the sentence on L356-357 discuss the insight that the classical approach for delineating basins is not appropriate and a preliminary check on the water balance is needed for karst system, especially if runoff coefficients are high. I am not sure of the novelty of this finding but this is a point that is noted in the title but then not mentioned again until the conclusions. The paper should be reframed with these contributions in mind. I will note again that I am not sure this will improve the novelty of the work but the conclusions are much more clearly stated as to the contribution of the work and it was unfortunate to wait until the end of the paper to understand the potential contributions of this work.**

**Authors' Reply:**

Thanks for the comment leading us to reframe the manuscript and to make everything more clear.

**Other Issues:**

Minor comments have been implemented in the revised manuscript. Additional modifications that have been added to the revised manuscript are as follows:

1. Based on the reviewer's comments, the Abstract is changed to accomplish their suggestions.

2. We improved Fig. 7, 8 and 9 in the revised manuscript by highlighting the proposed evaluation method (based on empirical conditional probability) demonstrating that the general classical scores are not enough to evaluate the models.

3. Based on the reviewer's comments, the Conclusions is modified also in the manuscript.


**Comment 1: The starting point is the non-correspondence of hydrographic basins and hydrogeological basins. This finding is a very basic one, everyone knows the difference and it is absolutely non a novelty for the scientific community. In their text, the Authors are presenting this issue as a novelty, instead of presenting the problem in the introduction chapter, using the relevant and abundant literature on this topic. So, the novelty of their manuscript has to be searched in the methods they applied to solve the problem. This is in my opinion the logical approach and I suggest them to completely rewrite the introduction focusing on the problem they want to analyse (how to take into account the overflow in river discharge due to external groundwater flow feeding your basin). By this way, they can easily highlight their findings, mainly related to the useful modeling and methodology they performed during the study.**

**Authors' Reply:**

We appreciate the reviewer for the valuable comment. While the non-correspondence of hydrographic and hydrogeological basins is not considered as the novelty, our paper focused on addressing the challenges associated with data scarcity in these basins. These challenges become more highlighted if we are going to investigate the effect of drought and climate change on these basins. We have proposed a reliable

approach that incorporates new methods and tools to accurately reproduce the water budget of such complex basins. We acknowledge that the original version of the manuscript did not effectively convey this message, and we have implemented significant revisions to the Introduction, Results, and Conclusions sections to clarify every thing. In the revised manuscript, these sections are highlighted in red to facilitate tracking and reviewing. We believe that these modifications will shed light on the significance of our work.

In particular, the introduction has been re-organized as follows:

1. We explained the general characteristics of the carbonate system.

2. We mentioned different approaches of taking account of external groundwater flux contributions to a basin together with the advantages and drawbacks of each approach. Furthermore, we described the challenges of the lumped modeling approaches versus the fully distributed ones. This allows the paper to be interesting for a broader audience.

3. We clarified the study objective in five main research questions (RQs) which is answered in different sections of the manuscript.

The research questions are as follows:

1. Can the complex response of carbonate catchments to precipitation be modeled with HDSys relying only upon streamflow and precipitation time series, aided by cross-correlation analysis?

2. What type of modeling solution is suitable for this task, and is a parsimonious modeling approach appropriate?

3. Are the classic goodness of fit scores enough to evaluate the reliability of the models?

4. What is the impact of external contributing areas on streamflow in catchments with fractured carbonate rocks? To what extent does this contributing area affect the total streamflow from small headwater catchments to the main outlet?

5. What is the role of storage in sustaining streamflow during the years with significant precipitation deficit in these types of catchments?

A proposal for the new Introduction can be found in the answer to the reviewer's #1 comments.

**Reviewer's Comment 2**
**The second concern is related to the English language. Too many sentences are too long, with secondary sentences included. The uses of commas is limited and this approach cannot be approved by international readers. Please rewrite the entire document using shorter and clear sentences: one concept, one phrase. I strongly suggest the support of a mother tongue for providing a successful review.**

**Authors' Reply:**

The manuscript has been reviewed by a native English speaker.

Herein below we try to answer the "Detailed comments". The other ones related to the typos and incorrect use of English words were all accepted and are not reported here.

**Detailed Comments**

1. **line 92: what do you intend with "linear" springs? Perhaps "streambed" springs?**

   **Authors:** Yes, thanks for the suggestion. We replaced the term and you can find it in cyan color.

2. **120: My?**

   **Authors:** This is a typo. It should be MU which is the acronym for Madonna dell'Uccelletto. It has been modified.

3. **line 160: you are in a karst domain, so a response in 3 days would be due to karst circuits. Please evaluate this possibility and if you exclude this possibility please explain why**

   **Authors:** According to Petitta et al. (2022), the continental deposits preserve this carbonate aquifer from the direct dissolution processes limiting the mature karst development in the saturated zones. Additionally, they demonstrated that the fast flow contributes to only a minor percentage of the discharge in this area and the groundwater circulation is mainly driven by fractures and fissures. That is why we have considered just 30 days as the response time of the carbonate catchment (as highlighted also in Nanni et al. (2020)). However, we have modified the manuscript to clarify that the study area is not considered as a full karst system (please see **Lines 106-108** in the revised text in cyan).

4. **caption of figure 3: " is still high" is qualitative evaluation, please specify the number (it seems that in this case is lower than 1, so why you think is high?)**

   **Authors:** The caption of Fig. 3 has been modified in the revised text and is highlighted in cyan: "(a1) Cumulative observed discharge at CSA versus cumulative precipitation recorded at the closest station to CSA; (a2) Coefficient time series computed by dividing the discharge at CSA by the precipitation time series recorded at different stations. (b1) Cumulative observed discharge at Visso versus cumulative precipitation related to a station close to Visso; (b2) Coefficient time series computed by dividing the discharge at Visso by the precipitation observed at several stations. (c1) Cumulative observed discharge at Ussita versus cumulative precipitation related to a station close to Ussita; (c2) Coefficient time series computed by dividing the discharge at Ussita by the precipitation observed at several stations. The 1:1 line is shown in green. For CSA and Ussita the runoff coefficient is about 4 and 1.5, respectively, and this value is around 1 at the outlet of the basin (Visso)"

5. **line 205: using the period 2017-2018, do you not have problems with the reaction to the earthquake? I read some papers indicating a long reaction in discharge in this zone**

   **Authors:** We have not considered the discharge records affected by the seismic sequences during Nov. 2016-Nov. 2018 (see Fig. 2 in the revised manuscript). Additionally, in the revised manuscript,

[Figure]

**Figure 1. This figure is Fig. 9 in the current version of manuscript:** (a) Simulated discharges at the Ussita outlet for the calibration period (2019-2021). The Uncertainty analysis results obtained by the ECP method for the simulated discharge are shown with the grey area (b) The flow duration curve for observed and modeled discharge are in red and blue, respectively.

115   we have modified the period of calibration for MU to Nov. 2018-2021. The model has been then validated at the outlet of the basin (Visso) during 2019-2021. In this regard, Fig. 1 in this rebuttal is the modified version of Fig. 9 in the manuscript and it has been replaced. **All the changes in section 5.1 (Lines 241-283) are specified in red.**

120

6. **line 297: this is the real core of your manuscript and this has to be highlighted both in the discussion and in the conclusion!**

125

**Authors:** Thanks for the useful comment. The content of section 5.2 and also the figures have been reanalyzed. Additionally, we highlighted the important role of storage in supporting the river discharge in the years with precipitation deficit. Furthermore, the recovery from dry years in these kinds of basins with long hydrological memory has been discussed. Section 5.2 has been totally
130   modified, **see Lines 320-368**.

7. **line 321: where is Pescara spring? Out of your study area? So why you includes this spring in your comments here? I suggest to cancel this reference**

**Authors:** We removed it from the text to avoid confusion.

135

8. **line 324: I did not find a "lack of clear recharge signal" in the reference you cited here. I suggest to cancel this part, not necessary and not included in your study area**
**Authors:** We removed it from the text accordingly.

9. **line 326: the sentence is not clear, please rephrase the concept. I know that aquifer recharge is EVER going to springs/river, producing discharge (not runoff)**
   **Authors:** We removed the sentence to avoid any ambiguity.

10. **line 328: you have not discussed the role of Karst, so I suggest to not include karst in the conclusion**
    **Authors:** Based on point 3 the basin cannot be considered fully karst so the text is modified accordingly.

11. **line 332: if you have karst, please discuss in the text, not in the conclusion**
    **Authors:** We reorganised the text to avoid ambiguity (see points 3 and 10). Thanks for mentioning this point.

12. **line 357: your findings are not based on isotopes neither in tracer tests, so why you added in the conclusion?**

    **Authors:** We have modified the Conclusions accordingly (**Lines 370-421**).

**Authors:**

ADDITIONAL modifications that have been added to the manuscript are as follows:

1. Based on the reviewer's comments, the abstract has been changed.

2. The Introduction has been rewritten.

3. We improved Fig. 7, 8, and 9 in the revised manuscript by highlighting the proposed evaluation method (based on empirical conditional probability) demonstrating that the general classical scores are not enough to evaluate the models.

---

## Author Response (AR2)

**Reply to the Comments of Reviewer 1: Second Revision**

**Rebuttal of the manuscript entitled "On understanding mountainous carbonate basins of the Mediterranean using parsimonious modeling solutions"**

We have sincerely appreciate the valuable comments provided by both the Associate Editor and the reviewer. We have carefully considered each of reviewer's comments and, hopefully, have taken the necessary steps to address them. In our revised version, we have provided detailed point-by-point responses and made corresponding modifications to the manuscript accordingly. **Reviewers' comments** are in boldface.

The colour of the text in the revised manuscript is as follows:

1. blue is used to specify the changes related to the **former** reviewer 1's comments.

2. cyan is used to specify the changes related to the **former** reviewer 2's comments.

3. red is showing the changes which refer to the **former**-common comments of both reviewers.

4. brown is used to specify the changes related to the **current** reviewer #1's comments (second round of revision).

**Reviewer's Comments:**

**The authors have made efforts to clarify the novelty of the discussion paper. While I still find that the use of only 3 stations with limited periods of overlapping record is difficult to make general conclusions about mountainous carbonate basins, my area of expertise is more focused on statistical methods in hydrology and I have to evaluate the discussion paper on this aspect. I leave the other aspect - the novelty of the modeling solution for carbonate basins - to hopefully be assessed by other reviewers.**

**Authors' Reply:**

We highly appreciate the reviewer's effort in critically evaluating our approach and providing a valuable feedback. We acknowledge that one of the limitations of our study is the limited number of stations with overlapping records. However, we have to consider that the scarcity of data is a common challenge even in today's data-rich era. Despite this limitation, we believe that by employing a physically based (albeit lumped) modeling approach and robust correlation analysis, the data shortage challenge could be still addressed. In fact, our findings align with the results of other studies which could further support the effectiveness of our proposed approach in mitigating the data limitation problem. In the main text, we have explicitly mentioned this as one of the limitations of our study and highlighted the potential benefits of utilizing a larger and more diverse dataset to gain a deeper understanding of catchment behavior.

Once again, we appreciate the feedback from the reviewer, which has allowed us to address this limita-
tion in our manuscript. We add the following sentences to the main text in lines 409-412:
"Overall, having more data with a longer period of overlapping records would be probably beneficial to
improve the simulation of such a complex basin behavior. Although one of the limitations of our study
is the limited number of stations with overlapping records, employing a physically based (albeit lumped)
modeling approach together with a robust correlation analysis could mitigate the data shortage issue."

**Reviewer's Comment 1:**

**There are still questions about the methodology that need to be clarified further before the dis-
cussion paper can be considered for final publication.**
**L132: Do you actually prove that the mean precipitation is constant in the red-shaded area of Figure
1? I may have missed this, but this assumption appears to be quite critical for the analysis and it is
unclear whether this is actually shown in this work.**

**Authors' Reply:**

First, we have to mention that the precipitation data has been spatially interpolated using the kriging
method by taking into account the gauges located outside of the basin, in addition to the gauges inside the
basin. Figure 1 (in the rebuttal) shows the location of the gauges applied for interpolating the precipitation
in the basin and over carbonate areas. This could guarantee a better representation of precipitation over
the red-shaded areas which has been applied as the input of the model for these areas. Second, as clari-
fied in the latest version of the manuscript and in our previous response, one of the primary objectives of
our study is to develop an efficient modeling approach for simulating the behavior of the carbonate sys-
tem, particularly in situations where data availability is limited. In Figure 1 (in the manuscript), we have
depicted two separate carbonate areas in red (upstream of CSA and Ussita) represented by two lumped
models. It is important to note that incorporating more spatial variability for carbonate areas will result
in increased model parameters, which could introduce additional uncertainties. Given the limited existing
data for calibrating the model parameters, we decided to simulate the groundwater contribution from the
carbonate area using only two additional lumped systems. This could satisfy our primary objective which
is to investigate the temporal variation of basin storage and its response to the drought at the three main
closure points, rather than capturing the overall spatial variability. Also, it should be considered that the
random and mixed nature of flow within the carbonate system tends to diminish the influence of individual
flow paths, allowing for a more simplified representation. The simulation scores together with the uncer-
tainty analysis show the adequacy of this choice in describing the system. We appreciate the reviewer's
attention to this aspect and assure them that our modeling approach, despite its simplifications, effectively
captures the temporal variation of basin storage and provides meaningful insights into the behavior of the
carbonate system under the given data limitations.
We add more details to clarify the issues mentioned in this comment (Line 184-192 in the text):

"In this study, the primary focus is on the temporal variation of the precipitation-recharge-discharge be-
havior of the AC water flowing from CSA and Ussita rather than the spatial variability of the carbonate
system's behavior. This allows us to specifically investigate the impact of single or multiple-year drought

[Figure]

**Figure 1.** We have taken into account the raingauges outside of the basin and close to the carbonate areas to better representation of precipitation over these areas. (The figure is associated with this rebuttal and it is not added to the main text)

events on the basin storage, as discussed in Section 5.2. So, a lumped modeling approach is more appropriate for providing more insights into the temporal system's response to drought conditions and its implications for basin storage. Furthermore, incorporating more spatial variability for the carbonate areas would result in an increased number of model parameters. This introduces additional uncertainty into the
85 model. Given the limited availability of data for calibrating these parameters, using two separate lumped systems has been considered as an efficient strategy for the modeling. The results will also demonstrate that the temporal behavior of the AC water and its response to drought events could be investigated properly by this modeling approach."

90

**Reviewer's Comment 2:**

Because of the limitations of using only 3 stations, it is very difficult to generalize any conclusions from this work. This is especially true when 1 of the 3 stations (the middle station on Figure 3) shows
95 a different relationship between cumulated precipitation and cumulated runoff. It does not appear to be discussed how this affects your assumptions or conclusions about groundwater influxes to the carbonate system. For example in L140, it is stated that "After understanding that the external contribution to the basin is significantly greater than that provided by the terrain analysis..." However, this is not true for this 1 station where cumulated precipitation is slightly less than cumulated runoff
100 in Figure 3. How is this explained? And why is this not mentioned?

**Authors' Reply:**

Please refer to the answers to the previous comments about using three hydrometric stations. We apologize for any confusion caused and not explicitly addressing the relationship between cumulated precipitation and cumulated runoff in different pannels of Figure 3.

The area of CSA, Ussita, and Visso increases in order, with CSA being the smallest and Visso being the largest which encompasses both CSA and Ussita. Figure 3 illustrates that the contribution from the carbonate (red) catchment decreases with increasing catchment size, which is a reasonable expectation. After Lines 131-138 in the text which is about the variation of runoff coefficient, Lines 138-141 are added as follows:

"The area of CSA, Ussita, and Visso increases in order, with CSA being the smallest and Visso being the largest which encompasses both CSA and Ussita. Fig. 3 illustrates that the contribution from the carbonate (red) catchment decreases with increasing catchment size, which is a reasonable expectation. To further clarification about the lower runoff coefficient of Visso, the readers could refer to Section 5.2 and Fig. 12."

However, it is important to consider this information in conjunction with Figure 12, which presents the water budget components. The water budget analysis in Figure 12(b) allows us to infer that there is a hidden subsurface flow in the catchment fed by the river, which causes sequential positive storage in the basin. This could indicate that the river basin feeds the groundwater system specifically between the CSA and Visso hydrometric stations. In Lines 363-369 in the main text, we have mentioned this feature of the basin as follows:

"For Visso (Fig. 12b) the storage differences remain positive for the period of interest, indicating a potentially infinite stored water accumulation over multiple years. Possible explanations are either that the basin feeds the groundwater system (not simulated by the model) between the CSA and Visso hydrometric stations (see also the observation discharge at CSA and Visso stations) or, considering the long-term memory of the basin, that ten years (2010-2021) is a relatively short period for observing storage changes. While both assumptions remain unanswered, the complexity of the system makes the first assumption plausible, however further investigations are needed to provide compelling evidence of this. Moreover, the first assumption could also justify the lower runoff coefficient at Visso station. "

To avoid any confusion, we also modified Figure 3 and sorted the panels of the figure (top to bottom) according to the order of subbasin area.

**Reviewer's Comment 3:**

**It is also confusing as a reader where the figures show 3 stations as CSA (which is not spelled out and figure captions should be stand-alone), Ussita, and Visso; however, the text continues to discuss a station MU and does not discuss Ussita. Are these the same station? Or was Ussita dropped for the analysis?**

**Authors' Reply:**

We apologize for the confusion caused by the inconsistency in the figure captions and the text about different station names.

To clarify, CSA, Madonna dell'Uccelletto (MU), and Visso are the three stations analyzed in our study. In fact, Ussita is the river where the MU station is located on that. So Ussita was not removed from the analysis; rather, it was sometimes mistakenly applied instead of MU in the text. The authors appreciate the reviewer for drawing our attention to this issue. It has been modified in the entire text and MU is applied referring to the station and Ussita is used to call the river basin in the current version of the manuscript. All the station name abbreviations have been already spelt out in Table 1. We also provided more clarification in Lines 101-104:

"The stations used in the study are Visso and Castelsantangelo (CSA) on the Nera River, and Madonna dell'Uccelletto (MU) on the Ussita River. To prevent any confusion, from this point forward in the text, "MU" will be used to refer to the station, while "Ussita" will be used to denote the river basin."

**Reviewer's Comment 4:**

**The new statistical approach that is introduced is difficult to understand.**

**Authors' Reply:**

The proposed method for evaluating the reliability of simulations involves calculating the empirical probability of any measured discharge conditional on the simulated value at the same time step.

To clarify the methodology, we have modified lines 227-239 of the text as follows:

"The process of computing the Empirical Conditional Probability (ECP) involves the following steps:

- Combining the observed discharge and the corresponding simulated values into a single dataset.

- Grouping the dataset into $n$ classes (bins) according to the simulated discharge values. The quantile-based discretization method has been applied for binning data into different classes (see Figure 2 in this rebuttal).

- Computing the Empirical Cumulative Distribution Function (ECDF) for each class $j$ using the formula:

$$ECDF_j(Q) = \frac{1}{m_j} \sum_{i=1}^{m_j} I_{X_i < Q} \tag{1}$$

Here, $ECDF$ represents the empirical cumulative distribution function of the $j$-th class, $m_j$ is the number of measures in the group, $X_i$ denotes the $i$-th measure in the group, and

$$I_{X_i < Q} = \left\{ \begin{array}{ll} 1 & \text{if } X_i < Q \\ 0 & \text{otherwise} \end{array} \right. \tag{2}$$

– Computing the features of empirical distribution function (i.e., mode, maximum, minimum, and mean of the discharge) for each class.

From the ECDF, we can derive the empirical distribution functions for different classes (e.g., the one shown in Fig. 6(b) and 6c), which are assigned to each time step. "

and also Lines 248-252 are added as follows:

"The histograms obtained for different bins (e.g., Fig. 6 (b) and (c)) are dedicated to the time steps visualised as green dots and grey-shaded areas in Fig. 7, 8, and 9. The green dots and grey area illustrated in the figures provide an indicator of the reliability of the simulations, according to previous simulated and observed data. In particular, the disparity between the measured and the mode values (green dots) can be considered as a measure of this reliability. The complete estimation procedure is thoroughly documented in a specific Notebook, accessible in the supplementary material. "

**Given that you only have 3 stations and very limited data for calibration and validation - in fact, as noted, data were so limited for calibration and validation that it was not always possible to use all stations for these purposes (L186-190) - how can the new statistical method have been robustly validated on this dataset? More justification is needed as to how this small set of data was able to provide reliable empirical PDFs.**

Since the available data are hourly, the computed empirical probability distribution function for each class -even for the shortest time series (MU station) with 26281 data points- is based on a meaningful number of data points. For this aim, the number of classes is specified in such a way that a meaningful histogram is obtained for each class. However, we mentioned this point in Lines 252-255 of the text.
"It is important to highlight that the number of classes (bins) has been carefully chosen to ensure a meaningful histogram for each discharge class. Even for the shortest available dataset (at the MU station, which encompasses approximately 26,000 hourly data points), a reasonable number of samples are available for each discharge class"

**Minor comments:**

**There are numerous areas where the referencing has two parentheses. A complete review of the discussion paper needs to be completed to address this aspect. See L108 and L118 for 3 examples of this.**

It has been corrected.

[Figure]

**Figure 2.** The features of empirical distribution function (i.e., mode, maximum, minimum, and mean of the discharge) for each class (The figure is associated with this rebuttal and it is not added to the main text)

**L94-97: The sentence in L94 states the data were "provided by the Marche Region." This is not clear as a Region cannot provide the data. Later in L97, it is stated that the Marche Regional Authority provided some data. Is that what was meant in L94? Also, a reference is needed for the Marche Regional Authority data in L96.**

We have written "Marche Region Authority" instead of "Marche Region"

**L4: Should read: "climate change conditions"**
It has been corrected.

**L5-6: Should read "describe behavior in carbonate basins"**
It has been modified.

**L132: Should read: "the null"**
It has been corrected.

**Reply to the Comments of Reviewer 2: Second Revision**

**Rebuttal of the manuscript entitled "On understanding mountainous carbonate basins of the Mediterranean using parsimonious modeling solutions"**

**Reviewer's Comments:**

**The revision provided by the Authors accomplished with all comment provided after the first submission phase. I consider the manuscript acceptable for publication in HESS**

We sincerely thank the reviewer for dedicating time and effort to thoroughly review our manuscript. We greatly appreciate their valuable comments, which have enhanced the quality of our work. We are grateful for their positive evaluation and consideration of our manuscript for publication in HESS.

---

## Author Response (AR3)

Trento, November 7, 2023

Dear Editor,

We are delighted to submit the final version of our manuscript, along with all the necessary files for publication. We would like to express our gratitude to the Editor for their invaluable assistance and the Reviewer for their work. Their feedbacks greatly enhanced the quality of our paper compared to its initial draft.

We have taken utmost care during the uploading process and trust that all files have been successfully submitted without any errors. However, should there be any issues, we would appreciate your prompt notification.

Thank you for your time and consideration.

On behalf of the authors,

riccardo rigon